Osteology of Batrachuperus londongensis (Urodela, Hynobiidae): study of bony anatomy of a facultatively neotenic salamander from Mount Emei, Sichuan Province, China

Jiang Jian-ping 1
Jia Jia 2
Zhang Meihua 1
Gao Ke-Qin 2 kqgao@pku.edu.cn
1 Chengdu Institute of Biology, Chinese Academy of Sciences , Chengdu , China
2 School of Earth and Space Sciences, Peking University , Beijing , China
Koyabu Daisuke
Electronic publication date: 2018 Mar 28
Publication date: 2018
Volume: 6
Electronic Location ID: e4517
Received 2017 Oct 11; Accepted 2018 Feb 25
Copyright: © 2018 Jiang et al.
Copyright year: 2018
Copyright holder: Jiang et al.
License: This is an open access article distributed under the terms of the Creative Commons Attribution License, which permits unrestricted use, distribution, reproduction and adaptation in any medium and for any purpose provided that it is properly attributed. For attribution, the original author(s), title, publication source (PeerJ) and either DOI or URL of the article must be cited.
License URL: https://creativecommons.org/licenses/by/4.0/

Keywords: Batrachuperus londongensis, Hynobiid salamander, Facultative neoteny, Osteology, Character evolution

Funding: National Natural Science Foundation of China NSFC 41672003, 41702002 Strategic Priority Research Program, Chinese Academy of Sciences XDA19050201 National Key R & D Program of China 2017YFC0505202 This research was supported by the National Natural Science Foundation of China (NSFC 41672003, 41702002), Strategic Priority Research Program of the Chinese Academy of Sciences (XDA19050201) and National Key R & D Program of China (2017YFC0505202). The funders had no role in study design, data collection and analysis, decision to publish, preparation of the manuscript. There was no additional external funding received for this study. The funders had no role in study design, data collection and analysis, decision to publish, or preparation of the manuscript.

==============================
The Longdong Stream Salamander Batrachuperus londongensis, living in a mountain stream environment at Mt. Emei in Sichuan Province, China, represents a rare species that is facultatively neotenic in the family Hynobiidae. Although the species has been known to science for some 40 years since its initial discovery in the late 1970s, anatomical details of its osteology remain poorly understood and developmental information is still lacking for the species. This study (1) provides a detailed osteological account of B. londongensis based on micro-CT scanning and clearing and staining of multiple specimens from the type locality; (2) provides a discussion of intraspecific variation related to life-history differences; and (3) presents a discussion on limb features related to morphological evolution of limb patterns correlative with ecological adaptation to mountain stream environments. Osteological comparisons with congeneric species has led to recognition of several diagnostic features that are unique to B. londongensis, including: vomers widely separated from one another, lacking a midline contact; presence of uncommon perichondral ossification of the ascending process of the palatoquadrate as part of the suspensorium; and presence of a prominent posterodorsal process of the scapular blade, which serves as a ligamentous insertion of the levator muscle of the scapula. In addition, some but not all neotenic individuals retain the palatine as a discrete element, indicative of its delayed absorption after sexual maturity. Postmetamorphic and neotenic individuals are strikingly different in the complexity of hyobranchial structures. Neotenes display a high degree of ossification of hyobranchial elements, tend to increase ossification of both hypobranchial I and ceratobranchial I during aging, and retain fully ossified ceratobranchial III and IV; in contrast, these elements remain entirely cartilaginous or are totally lost by resorption in postmetamorphic individuals. In addition, all postmetamorphic forms display an inverted “T”-shaped basibranchial II, whereas neotenes show transformation from a “fork”-shaped to the “T”-shaped configuration after sexual maturity. B. londongensis displays a mosaic of apomorphic and plesiomorphic states in its limb ossifications: presence of a single centrale element in both the manus and pes is a derived condition in Hynobiidae and other families as well, whereas retention of a postminimus in the pes is obviously plesiomorphic within Urodela. Reduction in number of digits from five to four in the pes and possession of a cornified sheath covering the terminal phalanges are also derived features shared with some but not all mountain stream salamanders that are adapted to a similar type of environment.

Introduction

The family Hynobiidae includes 66–67 species in 9–11 genera (Fei et al., 2006; AmphibiaWeb, 2017; Frost, 2017). These are small to medium sized salamanders found primarily in Asia, although geographical range of one species (the Siberian Salamander Salamandrella keyserlingii) extends from Asia into European Russia (AmphibiaWeb, 2017; Frost, 2017). Phylogenetically, the Hynobiidae form the sister clade with Cryptobranchidae, the two families together classified in the suborder Cryptobranchoidea Dunn (1922). The Hynobiidae have long been viewed as a group of primitive salamanders, as they lack derived features such as fusion of the angular with the prearticular in the lower jaw, but retain plesiomorphic features, including external fertilization (Dunn, 1923; Noble, 1931), a large number of chromosomes (2n = 40–78), and the presence of microchromosomes (Morescalchi, 1973, 1975; Edwards, 1976; Sessions, 2008).

Based on the fossil record from northern China, the evolutionary history of the Hynobiidae can be traced back to Aptian time (∼125 Ma) during the Early Cretaceous (Chen & Gao, 2009; Gao, Chen & Jia, 2013; Jia & Gao, 2016b). Several extinct taxa from China including Liaoxitriton and Nuominerpeton are apparently stem-group hynobiids, as evidenced by their possession of derived features shared with extant hynobiids, including: transverse and arched vomerine tooth rows; a deeply notched posterolateral border of the vomer for choana; and an optic foramen opening at the notched posterior border of orbitosphenoid (Chen & Gao, 2009; Gao, Chen & Jia, 2013; Jia & Gao, 2016a). The fossil record indicates that the split of the hynobiid from cryptobranchid clade seems to be a phylogenetic event that had taken place no later than the Aptian time (∼125 Ma). Recent analyses of the nuclear exon and mitochondrial genome estimate the Cryptobranchidae–Hynobiidae split as ∼150 Ma and the origin of crown-group hynobiids as ∼125 Ma (Zheng et al., 2011: fig. 3). A more recent analysis of nuclear genes, however, yielded an estimate of ∼157 Ma for the split and ∼135 Ma for the origin of crown-group hynobiids (Chen et al., 2015: fig. 2).

Most hynobiids are terrestrial in the adult stage, but species in several genera (Batrachuperus, Liua, Pachyhynobius, Ranodon, and Paradactylodon) are mostly mountain stream dwellers that can be found in water all year around (Reilly, 1983; Fei et al., 2006; Sparreboom, 2014; AmphibiaWeb, 2017). In terms of life-history features, most hynobiids have a biphasic life cycle, with a gilled aquatic larval stage going through metamorphosis to a postmetamorphic adult stage. The exception is the Japanese species Hynobius retardatus, which has been reported as partly neotenic as a population variant from Lake Kuttarush in Hokkaido, with some specimens reaching a total length (TL) of 150 mm while retaining gills (Sasaki, 1924). However, this population now appears to be extinct, with no osteological description of the population ever undertaken (Inukai, 1930–1932; Wakahara, 1996). Accordingly, Batrachuperus londongensis Liu & Tian, 1978, commonly known as the Longdong Stream Salamander, appears to be the only extant hynobiid species that has been documented as facultatively neotenic (Fei et al., 2006). Osteological study of this unusual hynobiid is bound to be significant for understanding patterns of ossification of the cranium and postcranium, and for reconstruction of the evolution of hynobiid salamanders more generally. Furthermore, such an osteological study is in urgent need because survival of the species at the type locality Longdong Stream is facing a severe threat from habitat destruction and illegal collection (see below).

Within Hynobiidae, the genus Batrachuperus includes six or seven species (Fei et al., 2006; Fu & Zeng, 2008; Fei, Ye & Jiang, 2010) that are all endemic to western China. The genus consists of the type species Batrachuperus pinchonii (David, 1872 (1871)) plus five other species (B. londongensis, Batrachuperus tibetanus, Batrachuperus yenyuanensis, Batrachuperus taibaiensis, and Batrachuperus karlschmidti; see Fei et al., 2006; Fu & Zeng, 2008). Another nominal taxon, Batrachuperus cochranae Liu, 1950, was synonymized with B. pinchonii based on molecular evidence (Fu & Zeng, 2008), but the validity of the name is still in dispute (Fei, Ye & Jiang, 2010, 2012). All of these are aquatic forms that inhabit mountain stream and/or plateau pond environments, at elevations ranging from 1,200 to 4,400 m above sea level (Fei et al., 2006; Fei, Ye & Jiang, 2010). In terms of conservation status, most of these species are currently vulnerable (Jiang et al., 2016). No fossil record has been found for Batrachuperus or its sister clade (Liua + Pseudohynobius). Based on molecular data alone, Batrachuperus was estimated as having originated in the late Miocene (∼7.3 Ma) in a recent analysis using 29 nuclear genes (Chen et al., 2015), a much younger date than the estimation of ∼24.3 Ma based on the complete mitochondrial genome (Zhang et al., 2006) or that of 20–30 Ma based on complete mitochondrial genome and three nuclear genes (Zheng et al., 2011).

Batrachuperus is one of the several hynobiid genera (Batrachuperus, Liua, Pachyhynobius, Paradactylodon, Pseudohynobius) that have received little study of their developmental osteology (Rose, 2003); moreover, anatomical details remain poorly known for these taxa. For the genus Batrachuperus, brief osteological accounts of the type species, B. pinchonii can be found in the literature (Zhang et al., 2009; Xiong et al., 2013a); however, adequate osteological description of B. londongensis is still lacking, although the species has been known to science for almost 40 years since its discovery in the late 1970s. In regard to phylogenetic relationships, few adequate data can be found in the literature on osteological details of the species. Attempting to improve this awkward situation, our study provides a detailed osteological account of B. londongensis and a morphological comparison in relation to life-history differences within this species; in addition, we present a discussion on diagnosis of the species based on bony features, patterns of ossification of the hyobranchium and limb ossification. Phylogenetic analysis will be performed in future research when necessary data becomes available for other congeneric species.

Materials and Methods

A total of 12 specimens of B. londongensis were used in this study, including both juvenile (1) and adults (11), and both postmetamorphic (4) and neotenic (8) individuals (Table 1). The material used includes both dry skeletons and fluid-preserved (10% formaldehyde) specimens. In addition, cleared and stained specimens (CIB 14499, 14504) allowed observation of both bony and cartilaginous structures. All of these specimens are deposited in the Chengdu Institute of Biology (CIB), Chinese Academy of Sciences, Chengdu, China. For purpose of comparison, several specimens belonging to congeneric species and other hynobiid species were used in this study (see Table S1).

Table 1 Measurements of specimens used in this study.

Catalogue number	Field number	Male/Female	External gill slits	Total length	Snout–pelvic length	Skull length	Skull width	
CIB 14500	川 I00202	Juvenile	Absent	110.07	60.24	16.84	12.82	
CIB 14504	65I0012	♀	Absent	220.15	117.51	26.96	21.28	
CIB 14507	620593	♂	Absent	225.35	120.32	34.08	25.69	
CIB 14509	638866	♀	Absent	219.99	113.15	28.19	23.08	
CIB 14380	65I0013	♂	Present	265.00	129.00	27.50	23.80	
CIB14381	638900	♀	Present	221.94	121.00	22.70	20.00	
CIB 14482	99I0511	♀	Present	178.28	98.10	25.94	17.38	
CIB 14484	99I 0517	♂	Present	189.05	97.93	26.73	18.35	
CIB 14485	99I0520	♀	Present	179.39	93.92	25.35	17.77	
CIB 14487	99I0513	♂	Absent	164.09	89.27	24.38	17.51	
CIB 14499	川I00196	♀	Present	214.76	116.96	27.85	21.31	
Note:

Measurements in mm.

Total length refers to the measurement between the tip of the snout and the posterior extremity of the tail; snout–pelvic length (SPL) is the measurement from the tip of the snout to the posterior end of the pelvis (marked by the anus in fluid-preserved specimens and by the haemal arch of the last caudosacral in dry skeletons). Skull length refers to the maximum dimension from the tip of the snout to the posterior end of the occipital condyles; skull width is the distance measured between the cranio-mandibular joints. All measurements are in millimeters (mm). Anatomical nomenclature follows Francis (1934), Reilly & Lauder (1988), and Shubin & Wake (2003), with the exceptions as noted in several cases.

The type specimen of B. londongensis is a large male, with a TL of 265 mm and a SPL of 129 mm (Fig. 1). The specimen carries a field number CIB 65I0013 as in previous publications (Fei, Ye & Tian, 1983; Fei et al., 2006), but it has been formally catalogued as CIB 65I0013/14380 in specimen collections at the CIB (Fig. 1). This specimen represents a neotenic form as it retains gill slits and a larval type of hyobranchium (Fei, Ye & Tian, 1983; Fei et al., 2006). The specimen was collected on March 23, 1965 from the type locality, Longdong Stream at Mount Emei (Emeishan), at an elevation of 1,300 m above sea level as has been documented in the literature (Liu & Tian, 1978; Fei, Ye & Tian, 1983). Subsequent collections of specimens were made in 2002, 2006, 2014, and 2016 from the same mountain stream by field crews from the CIB.

Figure 1 Photographs of ethanol-preserved specimens of Batrachuperus londongensis in dorsal view (A) holotype CIB 65I0013/14380; (B) referred specimen CIB 14381; (C) referred specimen CIB 14504.

All topotypic specimens were collected from the type locality Longdong Stream, Mt. Emei, Sichuan Province, China.

Selected specimens from the CIB collections (Table 1), including the holotype (CIB 65I0013/14380), were CT scanned using a high-resolution X-ray scanner (Quantum GX micro-CT Imaging System, PerkinElmer®, Waltham, USA) at the CIB, Chinese Academy of Sciences. These specimens were scanned along the coronal axis at an image resolution of 2,000 × 2,000. Comparative specimens from the FMNH collections were CT scanned at the University of Chicago (PaleoCT Lab) using a GE Phoenix v/tome/x 240kv/180kv scanner (Boston, USA). Segmentation and three-dimensional reconstruction of the CT images were made by using VG Studio Max 2.2 (Volume Graphics, Heidelberg, Germany).

Two specimens (CIB 14504, 14499) were whole-mount cleared and double stained after being scanned. The clearing and staining procedures followed the protocols of Hanken & Wassersug (1981). Cartilaginous elements were stained in blue using Alcian Blue 8GX and bony structures were stained in red using Alizarin Red, and then were cleared in glycerin KOH solution. Line drawings of skeletal structures were prepared using Adobe Photoshop® CS6 software, and are presented as text figures and supplementary figures (Supporting Information).

Results

Systematics and description

Order Urodela Duméril, 1806

Suborder Cryptobranchoidea Dunn, 1922

Family Hynobiidae Cope, 1859

Genus Batrachuperus Boulenger, 1878

Species Batrachuperus londongensis Liu & Tian, 1978

Holotype. CIB 65I0013/14380, a male adult with a TL of 265 mm, a snout-vent length of 129 mm. The holotype is a neotenic individual, as it is sexually mature but retains gill slits and a larval type of hyobranchium.

Type locality. Longdong Stream (N29°34′42.85″/E103°17′5.61″), at an elevation of 1,300 m above sea level, Mt. Emei (Emeishan), Sichuan Province, China.

Known distribution and habitat. The species was previously thought to be known only from Longdong Stream, a mountain stream associated with the Longdong Cave at 1,200–1,300 m above sea level (Fei et al., 2006); however, a molecular systematics study has shown that populations previously identified as B. pinchonii from the nearby Mt. Wawushan in Hongya County and Mt. Nibashan in Hanyuan County may pertain to B. londongensis (Fu & Zeng, 2008). Based on field observations at the type locality, live individuals of the Longdong Stream Salamander often hide under rock cover in the mountain stream, and they feed on fresh-water shrimp, aquatic insects and insect larvae (Fei, Ye & Jiang, 2010).

Diagnosis. Batrachuperus londongensis can be distinguished from the type species B. pinchonii and all other congeneric species in having a unique combination of the following osteological features: alary process of premaxilla barely contributing to border of anterodorsal fenestra (shared with B. tibetanus); suture between nasal and frontal located at the level of anterior border of orbit; frontal extending far posteriorly, terminating at the level of otic process of pterygoid; vomers widely separated from one another, lacking a midline contact (unique); vomerine teeth four to eight in number, arranged in a straight line that is strongly oblique and nearly vertical in orientation (unique); ascending process of palatoquadrate ossified as a short pillar propping lateral edge of parietal behind orbit (shared with B. karlschmidti and B. taibaiensis); radial loops stemming from basibranchial I not crossing one another; presacral vertebrae 18 in number; scapular blade bearing prominent posterodorsal process, onto which a levator muscle of the scapula inserts (unique); ischial plate is penetrated by a small nerve foramen, and the ischial spine is clearly more elongated than in other species.

Taxonomic remarks. In the literature, the species epithet has been confusingly spelled as B. londongensis (Liu & Tian, 1978; Liu et al., 1978; Fu et al., 2001; Wu & Xie, 2004; Fei et al., 2006; Fei, Ye & Jiang, 2010, 2012) and Batrachuperus longdongensis (Liu & Tian, 1983 in Fei, Ye & Tian, 1983; Fei & Ye, 1984; Song et al., 2001). Judging from the published literature regarding the two available names differing in spelling, it is clear that the former is valid and the latter invalid owing to its status as a “nomen nudum” (Fei et al., 2006).

The validity of the species name B. londongensis was established by Liu & Tian (1978) in Liu et al. (1978). The year before this publication, the same authors (Liu & Tian, 1977) first published the species name spelled as “Batrachuperus longdongensis” in a systematic checklist of Chinese amphibians (Liu & Tian, 1977) without a type designation, description, or illustration (see ICZN, 1999: art. 13); for that reason, the name “Batrachuperus longdongensis” Liu & Tian, 1977 is considered a nomen nudum (Fei et al., 2006). Unfortunately, Liu & Tian (1983 in Fei, Ye & Tian, 1983) “republished” the invalid name “Batrachuperus longdongensis” and labeled it as a new species. Although the latter publication provided some detailed information on the species, “Batrachuperus longdongensis” Liu & Tian, 1983 cannot be considered as the original spelling, but is a junior homonym of the nomen nudum “Batrachuperus longdongensis” Liu & Tian 1977. Fei et al. (2006) evidently chose to adopt B. londongensis as the original spelling in a valid publication by Liu & Tian (1978). Moreover, the species name B. londongensis is in prevailing usage in the current literature (Fei et al., 2006; Fei & Ye, 2017; Fu & Zeng, 2008; Fei, Ye & Jiang, 2010, 2012; AmphibiaWeb, 2017; Frost, 2017; IUCN, 2016).

External morphology

Formalin-preserved specimens provide information on external morphology except for coloration. Male adults of B. londongensis have been measured at 190–265 mm in TL, and female adults at 183–232 mm (Fei et al., 2006). Labial folds are well developed, partly covering the lower lips; gular folds are arched posteriorly to demarcate the head from the trunk. Neotenes have one to four pairs of gill slits, whereas in postmetamorphic individuals the gill slits are closed.

The presence or absence of eyelids and ring-shaped scleral cartilages in the eye are both ontogenetically and ecologically significant characters. Movable eyelids are normally developed at metamorphosis for adult life on land, but are often lacking in larvae and in those adults that are obligate neotenes (Duellman & Trueb, 1986). However, the ring-shaped scleral cartilage is partly resorbed at metamorphosis in Hynobius and Onychodactylus (Okajima & Tsuaki, 1921), while the extent of resorption of the scleral cartilage in other hynobiids is largely unknown (Rose, 2003). Our observations indicate that B. londongensis has movable eyelids but lacks scleral cartilages in both neotenes and postmetamorphic individuals. Although developmental information is still unavailable, we infer that it is likely that scleral cartilages are fully resorbed at metamorphosis or at sexual maturity as a normal developmental pattern as in other hynobiids. Because the species is facultatively rather than obligately neotenic, possession of movable eyelids may be indicative of their all-season aquatic life style being a secondary adaptation.

The trunk region is more or less cylindrical, having 12–14 costal grooves on each side along the trunk. A dorsal vertebral groove runs between the head and the base of the tail. The tail is one-half or slightly longer than one-half of TL. The tail has a cylindrical base, but becomes bilaterally compressed posteriorly, with evident dorsal and ventral fin folds extending distally. The end of the tail displays a rounded outline in lateral view (Fei & Ye, 2001). Toes in the hind limb are reduced to four (Fei et al., 2006; Fei & Ye, 2001), although occasional developmental anomalies are found in some individuals (see below). Terminal digits are claw-like, covered with a black cornified epidermal sheath.

Dermal skull roof

The skull roof is flat, longer than wide as in other congeneric species (Fei et al., 2006). The snout follows the contour of the maxillary arc: most specimens display a squarish outline, in contrast to the rounded snout in other congeneric species (e.g., B. pinchonii, B. karlschmidti, B. tibetanus). The maximum width of the snout across the anterior borders of orbits is much narrower than the maximum width of the skull at the cranio-mandibular joints, thus giving a more or less trapezoidal outline of the skull roof in dorsal view (Fig. 2; Video S1). This outline differs from that in B. pinchonii, which has a more rounded snout than B. londongensis, and displays significant differences in the maximum width of the snout and at the back of the skull (Fig. 2).

Figure 2 Skull roof of Batrachuperus londongensis (A) holotype CIB 65I0013/14380; (B) CIB 143481; (C) CIB 14482.

Note Specimens CIB 14381 (B) and CIB 14482 in (C) displaying display an incomplete resorption of palatine. Anatomical abbreviations used in this and other figures see list in Materials and Methods.

The paired premaxillae contact each other medially to close the anterior border of a large anterodorsal fenestra (De Beer, 1937: cavum internasale) in the skull roof and an anteromedial fenestra in the palate. The pars dentalis as the tooth-bearing part of the premaxilla forms the anterior wall of the snout between the external nares, but its lateral extension articulates with the maxilla to form the medial half of the ventral border of the narial opening. The anterior surface of the premaxilla is penetrated by several small foramina. The pars dorsalis (alary process) of the premaxilla ascends from the midlength of the pars dentalis, with the entire spine of the process set in a groove on the anterodorsal surface of the nasal bone; therefore, the alary process essentially contributes no part to the lateral border of the anterodorsal fenestra. A similar pattern is seen in B. tibetanus (FMNH 5901), but not in other species of the genus.

The paired nasals are strongly widened to display a dimension almost twice the width of the frontals, a plesiomorphic feature in urodeles as commonly seen in other hynobiids (Dunn, 1923; Gao & Shubin, 2012; Jia & Gao, 2016a). In dorsal view, the nasal is a large plate, irregular in shape, meeting its counter element along a midline suture. The nasal has an anterior process, which is medially notched for the large anterodorsal fenestra and laterally notched for the narial opening. A lateral process of the nasal is in limited contact with the lacrimal, because a large part of the latter bone overlaps the prefrontal (Fig. 2). The dorsal surface of the nasal is smooth, but is penetrated by several tiny foramina (foramen mediale nasi of Francis, 1934) that serve as passage of ultimate twigs of the mesial branch of the ophthalmicus profundus nerve (CN V1) as seen in Salamandra (Francis, 1934). The nasal posteriorly overlaps the frontal extensively, and laterally meets with the lacrimal and prefrontal. In dorsal view, the suture between the nasal and frontal is located at the level of the anterior borders of the orbits.

The paired frontals are strongly elongated posteriorly, with a straight or slightly sigmoid sutural contact along the midline. The paired elements occupy much of the interorbital area of the skull roof, but form only a small part of the medial border of the orbit, with a large part of the border furnished by the parietal. The frontal articulates with the nasal anteriorly and with the prefrontal anterolaterally. Immediately behind the posterior process of the prefrontal, the lateral border of the frontal curves downward to articulate with the orbitosphenoid; however, this contact is limited to only the anterior one-third of the orbitosphenoid, with the posterior two-thirds of the latter in contact with the parietal (Figs. 3 and 4). This short contact between the two elements is also seen in B. karlschmidti, whereas in all other congeneric species the prefrontal contacts over one-half of the dorsal margin of the orbitosphenoid (Figs. 3 and 4). Posteriorly, the frontal overlaps the parietal extensively, with a strongly elongate posterior process extending to the level of the medial (otic) process of the pterygoid. This strong posterior extension of the frontal differs significantly from all other congeneric species, in which the frontal is proportionally shorter and terminates roughly at the midlevel of the orbit (e.g., FMNH 49380: B. karlschmidti; FMNH 5901: B. tibetanus). Along the medial border of the orbit, the lateral margin of the frontal curves downward to meet the orbitosphenoid and the anterolateral process of the parietal.

Figure 3 Lateral view of the skull of Batrachuperus londongensis (A) CIB holotype CIB 65I0013/14380; (B) CIB 143481; (C) CIB 14482.

Specimens CIB 14381 (B) and CIB 14482 (C) display an incomplete resorption of palatine. All images with squamosal and quadrate removed to expose the otic capsule. Note the unusual perichondral ossification of the ascending process of palatoquadrate as a pillar between the parietal and pterygoid anterior to the prootic.

Figure 4 Palatal view of the skull of Batrachuperus londongensis (A) holotype CIB 65I0013/14380; (B) CIB 143481; (C) CIB 14482.

Specimens CIB 14381 (B) and CIB 14482 (C) display an incomplete resorption of palatine. Note all specimens show that the two vomers have no midline contact behind the anteromedial fenestra; note also delayed resorption of palatine in CIB 14482 (C).

The parietal is the largest element on the skull roof, but is extensively overlapped anteriorly by the posterior extension of the frontal. The paired parietals meet at a straight or slightly curved midline suture, and have parallel lateral borders forming most of the medial rim of the orbit. The main part of the parietal table, however, is expanded bilaterally to form robust lateral “boots” in articulation with the squamosals. The posterior part of the parietal table is rugose dorsally, and posteriorly bears a bony ridge that curves anteriorly to house a deep fossa (cervical epaxial fossa); both the bony ridge and fossa serve as insertion of the anterior epaxial muscles (Carroll & Holmes, 1980; Duellman & Trueb, 1986; Elwood & Cundall, 1994). Slightly anterior to the lateral boot of the parietal, the lateral surface of the parietal is pierced by a tiny foramen that serves as the passage of the trochlear nerve (CN IV; Gaupp, 1911) as clearly seen in several specimens (e.g., CIB 65I0013/14380, 14381, 14504, 14507, 14509). This foramen has been identified in all other congeneric species that we have examined (B. karlschmidti, B. taibaiensis, B. tibetanus, B. pinchonii, B. yenyuanensis) in this study and has also been reported in several other salamanders (e.g., Francis, 1934: Salamandra; Wake, 2001: Dicamptodon; AmphibiaTree, 2008: Ambystoma).

In both dorsal and lateral views, the parietal table sends an elongate anterolateral process pinched between the frontal and orbitosphenoid (Figs. 3 and 4). Anteriorly, the pointed end of this process terminates slightly anterior to the midlevel of the orbitosphenoid. At the posterior border of the orbit, the parietal develops a short triangular process, which is ventrally directed in contact with a distinct pillar-like bone that in turn is ventrally in contact with the pterygoid. This small pillar is identified as the perichondral ossification of the ascending process of the palatoquadrate, and the presence of such a distinct bony element in B. londongensis differs from all other congeneric species but B. karlschmidti and B. taibaiensis (see below).

The prefrontal is a robust element, set at an oblique position in front of the large orbit. The prefrontal meets the nasal and frontal medially; it underlies the lacrimal anteriorly and the facial process of the maxilla laterally. The posterolateral margin of the prefrontal is curved to form the anterodorsal rim of the orbit. The prefrontal has a well-developed posterior process that extends far posterior to the nasal-frontal suture. A similar pattern is seen in B. tibetanus, but this process is much shorter in other congeneric species (B. karlschmidti, B. taibaiensis, B. pinchonii, B. yenyuanensis). A prefrontal is normally present in most salamanders, but is absent in proteids, sirenids, and some but not all plethodontids (Trueb, 1993; Reilly & Altig, 1996; Rose, 2003).

The lacrimal is a narrow and slightly elongate plate, with its main part overlapping the prefrontal. The lacrimal is essentially rectangular, similar to that in B. tibetanus and B. yenyuanensis, but differing from that in other congeneric species, in which the lacrimal is triangular with a pointed anterior end (B. pinchonii and B. taibaiensis) or more or less “L”-shaped with a short process bending posteromedially (B. karlschmidti). In B. londongensis, the lacrimal has a limited contact medially with the nasal, but is in an extensive sutural articulation laterally with the facial process of the maxilla. The canal for the nasolacrimal duct (ductus nasolacrimalis) opens as a foramen posteriorly on the dorsal surface of the lacrimal (Fig. 2A), with the anterior foramen opening on the anteroventral side of the lacrimal. As a common feature seen in most of Batrachuperus species, the lacrimal extends anteriorly over the border of the external naris, but is posteriorly blocked by the prefrontal-maxillary contact from entering the border of the orbit (Fig. 2). In other hynobiids, the lacrimal variably enters the naris only (e.g., Salamandrella), enters the orbit only (e.g., Pachyhynobius, Paradactylodon, some but not all species in Hynobius), or enters both the naris and orbit (e.g., Liua, Onychodactylus, Pseudohynobius, Ranodon, some species of Hynobius).

The maxilla is extremely short, but massively built, laterally covering the facial area of the skull. The pars dorsalis of the maxilla is a robust process that widens between the narial opening and the orbit. The process has an extensive sutural articulation with the lacrimal but a limited contact with the prefrontal posterior to the lacrimal. The lateral wall of the maxilla is slightly convex, and in some specimens (e.g., CIB 14485, 14509) is penetrated by a small foramen anteriorly close to the narial rim. Ventral to the narial opening, the anteroventral process of the maxilla articulates with the premaxilla to form one-half of the ventral border of the naris. The posterior border of the pars dorsalis forms a part of the orbital rim, with the remaining part furnished by the prefrontal. On the inner side of the pars dorsalis, a deep groove leads to a small opening of the infraorbital canal, through which the superior alveolar branch of the trigeminal nerve (CN V2) and its associated blood vessels pass as in other salamanders (Francis, 1934). In both neotenic and postmetamorphic individuals, the short posteroventral process of the maxilla terminates at a level far anterior to the midlevel of the orbit. The short maxillary tooth row contains no more than 18 teeth. The tooth row terminates close to the posterior extremity of the maxilla.

The septomaxilla is a small bone exposed anteromedial to the narial border of the maxilla. A septomaxilla is present in all hynobiids, and is also present in some other groups of urodeles (Rose, 2003: ambystomatids, dicamptodontids, rhyacotritonids, and some but not all plethodontids). The absence of this element in several groups (sirenids, cryptobranchids, amphiumids, and proteids) cannot be simply explained as a “paedomorphic loss” (contra Duellman & Trueb, 1986), because the absent condition is also seen in metamorphosed salamandrids (Francis, 1934; Rose, 2003), let alone that the element is present in the Jurassic neotenic salamanders Beiyanerpeton and Qinglongtriton (Gao & Shubin, 2012; Jia & Gao, 2016b). In this study, all known specimens of B. londongensis consistently display the septomaxilla in both neotenic and postmetamorphic adults. Ontogenetically, the septomaxilla is ossified immediately before or during metamorphosis in Onychodactylus (Vassilieva, Poyarkov & Iizuka, 2013), Ranodon (Schmalhausen, 1968; Lebedkina, 2004), Salamandrella (Schmalhausen, 1958; Lebedkina, 1964; Regel, 1970), and Hynobius (Vassilieva et al., 2015).

Suspensorium

The squamosal is roughly T-shaped with its widened proximal end in articulation with the lateral “boot” of the parietal and a transverse bar extending ventrolaterally over the quadrate (Fig. S1). In dorsal view, the transverse bar is set at right angles in relation to the sagittal plane of the skull; in occipital view, however, the transverse bar is set in an oblique position, sloping ventrolaterally at a 45° angle towards the cranio-mandibular joint (Fig. S2). At the proximal end, a blunt otic process projects anteriorly to articulate with the prootic, and a triangular process projects posteriorly with its tip in contact with the opisthotic–occipital complex (Fig. 2). The squamosal is slightly convex dorsally, but concave ventrally to embrace a large part of the quadrate.

The quadrate, as a principal element of the suspensorium, extends transversely beneath the squamosal. In contrast to the squamosal, the quadrate has a widened lateral end and a medial process extending towards the stapes (columella). In dorsal view, the lateral end of the quadrate is exposed beyond the distal end of the squamosal, where the quadrate thickens ventrally to form a cartilage-lined saddle in articulation with the articular in the mandible. From the distal end, a short and flat process overlaps the posterolateral process of the pterygoid. As revealed in CT images of several specimens (e.g., CIB 65I0013/14380, 14381, 14507), a tiny quadrate foramen pierces the base of the pterygoid process as passage of a branch of the posterior condylar artery and vein as generally seen in other tetrapods (Olson, 1966). A quadrate foramen has been identified in the Early Cretaceous hynobiid-like salamander Nuominerpeton (Jia & Gao, 2016a), whereas the occurrence of the foramen in extant hynobiids needs to be investigated thoroughly to understand its phylogenetic significance. The ascending process of the quadrate extends beneath the squamosal, and has a ligamentous connection with the stapes (see description of the stapes below). Ossification of the quadrate occurs in all salamanders except sirenids (Rose, 2003). Ontogenetic ossification of the quadrate in different hynobiid species can be variably before (Suzuki, 1932; Lebedkina, 2004; Jia & Gao, 2016a), during (Lebedkina, 1964), or immediately after metamorphosis (Vassilieva et al., 2015).

The pterygoid is triradiate in ventral view (Fig. 4; Fig. S3). The anterolateral (palatal) process is elongated, extending to a point slightly anterior to the midlevel of the orbit; thus, the ligamentous connection with the maxilla is extremely short. As a common pattern in other salamanders, all hynobiids have a ligamentous connection between the maxilla and pterygoid, with the exception of Pachyhynobius, in which a bony contact between the two elements is established (Fei et al., 2006). In B. londongensis, the ventral surface of the anterolateral process is smooth, but the dorsal surface is grooved to receive a slender rod of cartilage (processus pterygoideus of the palatoquadrate), which extends along the entire length of the anterolateral process of the pterygoid.

The posterolateral (quadrate) process of the pterygoid is slightly shorter than the anterolateral process. A large part of the former process underlays the quadrate to reinforce the cranio-mandibular joint. In addition to serving as a part of the suspensorium, the pterygoid also serves as insertion of adductor muscles, including the pterygoideus head of the adductor mandibulae internus (Carroll & Holmes, 1980). The medial (otic) process of the pterygoid is short, but turns dorsally to meet a small pillar (ossified ascending process of the palatoquadrate); thus, the process has no bony contact with the parasphenoid. Lack of a synostotic contact between the pterygoid and parasphenoid is a common pattern in all other Batrachuperus species.

As mentioned above, it is of special interest to note the presence of a short pillar upholding the parietal immediately anterior to the prootic (Figs. 2 and 3). To our knowledge, this short pillar represents the perichondral ossification of the ascending process of the palatoquadrate (Trueb, 1993: metapterygoid; Rose, 2003: epipterygoid). The short pillar is set in a vertical or slightly oblique orientation, dorsally in articulation with a downward process of the parietal and ventrally with the upward medial process of the pterygoid. In other salamanders, the ascending process of the palatoquadrate normally remains unossified, whereas a “certain amount of perichondral ossification” of the process has been documented in Salamandra (Francis, 1934: 26) as part of the suspensorium.

In this study, all of the adult specimens of B. londongensis that we have examined consistently show the presence of this pillar in the same position, and the same contact patterns of the bone with the parietal and pterygoid can be seen in both dorsal and lateral views (Figs. 2 and 3). CIB 14500 is a postmetamorphic juvenile, which shows that this pillar is not yet ossified, as the individual also lacks ossification of the articular in the lower jaw (see below). However, B. karlschmidti (FMNH 49380) and B. taibaiensis (CIB 20040235) are the two congeneric species that show the presence of this bone and the same articulation patterns as in B. londongensis. In other hynobiids, ossification of the pillar as part of the suspensorium is also seen in Paradactylodon mustersi (FMNH 211936) and Pseudohynobius flavomaculatus (CIB 79I0107/17344). Whether this ossification represents a plesiomorphic feature with a homologous origin or a derived state independently acquired needs to be investigated in a phylogenetic analysis.

Palate and braincase

Formation of the palate involves the partes palatina of the premaxillae and maxillae, the vomers, and the anterior portion of the parasphenoid. The partes palatina of the premaxillae are the lingual shelves of the premaxillae that meet at the midline to close the anterior border of the large palatal fenestra (Fig. 4; Fig. S3). The fenestra (Schmalhausen, 1968: intervomerine cleft; Trueb, 1993: anteromedial fenestra) is the palatal opening for the intermaxillary gland (Schmalhausen, 1968). Laterally, the pars palatina of the maxilla contacts the vomer to complete the lateral rim of the palate.

In some neotenic individuals (e.g., CIB 14381, 14482), the ossified palatine is retained as a discrete element either in articulation with or free from the anterior process of the pterygoid on both sides of the palate (Figs. 2–4). In these specimens, the toothless palatine occurs as a small plate or a slender bar. The palatine is entirely resorbed in all other hynobiids in the adult stage as is commonly seen in other salamanders except sirenids (Worthington & Wake, 1971; Smirnov & Vassilieva, 2002; Rose, 2003); therefore, the retention of this element as an ossification in some but not all specimens in B. londongensis can be interpreted as an ontogenetic feature. This interpretation is supported by the hynobiid-like fossil salamander Nuominerpeton aquilonaris of Early Cretaceous age from China, which displays normal resorption of the palatine at metamorphosis (Jia & Gao, 2016a). Furthermore, none of the specimens of B. londongensis that we examined have shown a separate palatine at a postmetamorphic stage, and large neotenic individuals (e.g., CIB 65I0013/14380, 14484, 14485) also lack this element. Because those specimens having a discrete palatine are apparently adult individuals, ontogenetic resorption of the element in these individuals seems to be delayed until after sexual maturity.

The vomer is a large plate, irregular in shape, slightly longer than wide. The two vomers are widely separated from one another, thus allowing the parasphenoid to enter the posterior border of the anteromedial fenestra (Fig. 4; Fig. S3). This pattern is consistently observed in all specimens, and thus is recognized in this study as a unique feature different from all other species in Batrachuperus and other hynobiids as well. Jömann, Clemen & Greven (2005: fig. 33) figured Ranodon sibiricus at the adult stage as showing no midline contact of the vomers, but this was probably based on juvenile or subadult specimens (Jömann, Clemen & Greven, 2005: Table 1: TL ≤ 150 mm) as the nasals are widely separated (Jömann, Clemen & Greven, 2005: fig. 32) in contrast to the condition in fully grown adults (maximum TL = 250 mm) as figured in Fei et al. (2006: fig. 67). In B. londongensis, the medial border of the vomer is slightly concave, forming the entire lateral margin of the anteromedial fenestra (Fig. 4; Fig. S3). Posterolaterally, the vomerine plate is notched for the choana, with a small triangular process projecting towards the orbit to embrace the notched choana. However, a retrochoanal (postchoanal) process is absent, as a laterally directed process to block the posterior border of the choana is clearly lacking. Another projection, medially attached to the parasphenoid, represents the posterior process of the vomer (Fig. 4; Fig. S3).

A large part of the ventral surface of the vomer is smooth, except for a small tooth-bearing area medial to the choana. The vomerine teeth are four to eight in number, arranged in a linear fashion to form a straight row. The two vomerine tooth rows are widely separate but slightly oblique, convergent anteriorly and divergent posteriorly. This arrangement differs from most other congeneric species: in them the vomerine tooth row is more or less arched and parallel to the maxillary tooth row. Close to the premaxilla-vomer suture, the vomer is penetrated by two to three small foramina observed in several specimens (CIB 65I0013/14380, 14381, 14504, 14509; Fig. S4), probably for passage of the ramus ventralis of the trigeminal nerve (CN V1) and the ramus palatinus of the facial nerve (CN VII) to supply the mucous membranes of the mouth, as in other salamanders (Francis, 1934: Salamandra salamandra; Cloete, 1961: Rhyacotriton olympicus).

The parasphenoid is a dermal bone that forms a large median plate, roughly rectangular in outline, with straight lateral borders parallel to one another. The anterior process of the parasphenoid contributes to part of the palate and closes the posterior border of the anteromedial fenestra (Fig. 4; Fig. S3). The anterolateral part of the parasphenoid bears a flat facet, where part of the vomer posterolateral to the anteromedial fenestra is attached. The lateral edge of the parasphenoid curves dorsally, with a narrow groove in articulation with the ventral edge of the orbitosphenoid. Posteriorly, the basal plate of the parasphenoid is widened to form the lateral ala that floors the otic capsule. Close to the anterolateral border of the basal plate, a pair of internal carotid foramina penetrates the parasphenoid as passage of the internal carotid arteries. As observed in several specimens (CIB 65I0013/14380, 14381, 14482, 14504, 14507, 14509), more than one foramen may occur on either side of the parasphenoid, possibly indicating minor branches of the artery. The lateral ala has no bony contact with the otic process of the pterygoid, but has an articulation with the prootic (Fig. 4; Fig. S3). The posterior end of the parasphenoid forms a blunt posteromedian process, which furnishes the ventral rim of the foramen magnum.

The orbitosphenoid (sphenethmoid) as an endochondral bone is a rectangular plate that covers a large part of the anterolateral wall of the braincase. The bony plate dorsally articulates with the frontal and parietal, but in some specimens (e.g., CIB 65I0013/14380) it also articulates with the prefrontal (Figs. 3 and 4; Fig. S1). The entire ventral edge of the orbitosphenoid meets the parasphenoid. The anterior edge of the orbitosphenoid is vertical, forming a straight border of the orbitonasal fenestra; the posterior border, however, is deeply notched for the optic foramen, carrying the optic nerve and its associated vessels (Francis, 1934; Fox, 1959). There is no foramen in the orbitosphenoid as passage of the oculomotor nerve (CN III); instead, this nerve passes through the optic-oculomotor commissure, which is covered by cartilaginous tissues posterior to the orbitosphenoid. This pattern of cranial nerves exiting through the anterolateral wall of the braincase has been recognized as a diagnostic feature of the Hynobiidae (Jia & Gao, 2016a), differing from the sister group Cryptobranchidae, in which the optic foramen is fully surrounded by the bony rim of the orbitosphenoid.

A separate operculum is absent in B. londongensis, as is common in most of other hynobiids but Hynobius, in which the operculum is present as a separate element (Monath, 1965; Rose, 2003). An operculum is figured as present in Liua shihi (Zhang, 1985) and Ranodon sibiricus (Jömann, Clemen & Greven, 2005: fig. 34); however, the element that is labeled in both cases seems to be the stapes as it has a rounded footplate fused with the stylus.

No operculum, bony or cartilaginous, is identified in any of the specimens. The stapes (columella) has a massive and disc-like footplate covering the foramen ovalis. The stylus is extremely short, fused with the footplate proximally but with a small facet at its obtuse distal end, where the ligamentum squamoso-columellare attaches; the ligament connects the stapes with the quadrate and squamosal (Kingsbury & Reed, 1909). A stapedial foramen is unexpectedly present on both sides in two specimens (CIB 65I0013/14380, 14381; both neotenic individuals), penetrating the stylus horizontally (Fig. S2). In two other specimens (CIB 14507, 14509; both postmetamorphic individuals), a foramen penetrates left and right stapes, as well. However, none of the other specimens, neotenic or postmetamorphic, show the presence of the foramen. Therefore, the stapedial artery and/or the hyomandibular trunk of the facial nerve (CN VII) in this species can variably pass through the stapedial foramen or below the short stylus. A similar condition is known for Ranodon sibiricus: Schmalhausen (1968) described both present and absent conditions of the stapedial foramen in specimens observed by him. Although Trueb & Cloutier (1991) interpreted the absence of a stapedial foramen as a batrachian synapomorphy, its presence in some salamanders is more likely to be a plesiomorphic feature for Urodela because it also occurs in the stem caudate Karaurus (Ivachnenko, 1978; Estes, 1981). Furthermore, the foramen is found in the Late Jurassic salamandroids Beiyanerpeton and Qinglongtriton from China (Jia & Gao, 2016b). The discovery of the presence of this foramen in some but not all postmetamorphic individuals in B. londongensis casts more doubt on our understanding of the evolution of this character.

The prootic is irregular in shape, covering the lateral wall of the anterior portion of the otic capsule. The element dorsally articulates with the parietal, but a small part of its dorsal rim underlays the squamosal as well. Ventrally, the prootic articulates with the lateral ala of the parasphenoid. In lateral view, the alary process (optic process) of the prootic is a small flap directed anterodorsally; the inferior process (basal process) of the prootic is more or less triangular, with its ventral edge in articulation with the parasphenoid (Fig. 3; Fig. S1). A large foramen faciale penetrates the lateral surface of the prootic as passage of the facial nerve (CN VII).

The opisthotic and exoccipital are fully fused to form an opisthotic–exoccipital complex covering the posterior wall of the braincase. The complex dorsally articulates with the parietal and ventrally with the parasphenoid; in dorsal view, it is swollen posterolaterally as the housing of the auditory capsule. The opisthotic portion of the complex borders the fenestra ovalis anteriorly, whereas its exoccipital part borders the foramen magnum posteromedially and bears the occipital condyle posteriorly in articulation with the cotyle of the atlas (Fig. S2). At the base of the occipital condyle, the complex in penetrated by the foramen post-oticum (Fig. S2) through which the combined glossopharyngeus-vagus nerves passes (CN IX + X; Francis, 1934). In all hynobiids the opisthotic–exoccipital complex is fused and the prootic free (Carroll & Holmes, 1980; contra Trueb, 1993: three separate elements), except for Onychodactylus, in which the prootic is fused to the opisthotic–exoccipital complex (Smirnov & Vassilieva, 2002; Vassilieva, Poyarkov & Iizuka, 2013). In occipital view, the foramen magnum is roughly triangular in outline. The dorsal rim of the foramen has a median gap where the cartilaginous tectum synoticum attaches below the parietals. The ventral border of the foramen magnum is largely formed by the posteromedian process of the parasphenoid.

Mandible

As in all cryptobranchoids (hynobiids and cryptobranchids), the lower jaw of B. londongensis consists of the dentary, prearticular, angular, and articular. In addition, a small posterior mental process seen in all specimens at the symphysis gives a clear indication of the co-ossification of a mentomeckelian element with the dentary (Fig. 5; Fig. S3).

Figure 5 Mandible of Batrachuperus londongensis right mandible of CIB 14482 in medial (A) and lateral (B) views; mandibular arch of CIB 14482 in dorsal (C), and ventral (D) views.

The dentary is the largest bone of the mandible, covering most of its lateral and ventral aspect. The lateral surface of the dentary is smooth, slightly swollen along the lower border but weakly concave along the tooth row. A large foramen is often located at a position slightly posterior to the midlevel of the dentary tooth row (Fig. 5). This foramen marks the passageway of the mandibularis externus nerve (CN V3), innervating both the skin and the muscle (M. intermandibularis) between the two rami of the lower jaw (Francis, 1934). Posteriorly, the dentary terminates at a robust process below the cranio-mandibular joint.

Posteroventral to the dentary, the angular bone is exposed in medial, ventral, and lateral views. As observed in both lateral and ventral views, the angular is slightly thickened posteriorly but narrows anteriorly, where it is wedged between the prearticular and the dentary (Fig. 5; Fig. S3). The pointed anterior process terminates at a level slightly posterior to the anterior extremity of the anterolateral process of the pterygoid, and the thickened posterior end forms the posteroventral extremity of the mandible. In medial view, a small angular foramen opens anteroventrally at the angular-prearticular suture, close to the anterior end of the angular and below the coronoid process of the prearticular.

The prearticular is a large element covering most of the medial aspect of the jaw. Anteriorly, the bone terminates at a point close to the mandibular symphysis. Posterodorsally, the prearticular rises to form a large ascending process (coronoid process) that serves as insertion of the adductor mandibulae internus. The process slopes downward posteriorly and continues as a ridge that attaches to the medial side of the articular at the glenoid fossa. Below and posteroventral to the coronoid process, the prearticular is penetrated by one or two foramina (inferior dental foramina) as passage of the inferior alveolar ramus of the facial nerve (CN VII) and the alveolar artery (Francis, 1934). There is often a third foramen that serves as passage of the same nerve and vessel opening more anteroventrally at the tip of the angular bone.

The articular is well ossified in adults as in other extant species and fossil relatives of hynobiids but is absent in cryptobranchids (Reese, 1906; Rose, 2003; Jia & Gao, 2016a). In comparison to the angular, the articular is slightly shorter but more stoutly built. In dorsal view, the anterior process of the articular extends between the prearticular and dentary, terminating at an obtuse tip medial to the coronoid process of the prearticular (Fig. 5). The slightly expanded posterior part of the articular is in articulation with the quadrate. As revealed from CT-scanned images, a tube-like canal runs horizontally through the articular as the passage of the ramus hyomandibularis of the facial nerve (CN VII); whether this pattern occurs in other congeneric species and other hynobiids needs to be thoroughly investigated. Ontogenetically, the articular ossifies slightly later than the septomaxilla after metamorphosis (Smirnov & Vassilieva, 2002; Rose, 2003; Lebedkina, 2004; Vassilieva, Poyarkov & Iizuka, 2013; Vassilieva et al., 2015); thus, presence of a bony articular is a clear indication of maturity in fossil and extant hynobiids (Jia & Gao, 2016a). In this study, CIB 14500 is a postmetamorphic juvenile (TL = 110 mm; SPL = 60.24 mm), which displays a typical postmetamorphic pattern of the hyobranchium but shows no ossification of the articular. Once again, comparison of this with other specimens indicates that the articular is ossified after metamorphosis in B. londongensis. This ossification sequence of the articular is similar with some other hynobiids, for example, Onychodactylus japonicus (Smirnov & Vassilieva, 2002) and Hynobius formosanus (Vassilieva et al., 2015), but differs from that in Ambystoma, in which the element is ossified during metamorphosis (Reilly, 1987).

Dentition

Tooth-bearing elements in B. londongensis include the premaxilla, maxilla, vomer, and dentary. The premaxilla carries seven to nine teeth; the tooth rows on both sides are slightly curved to form a broad arc, corresponding to the blunt snout. The maxilla carries 16–18 teeth, with the tooth row terminating at the posterior extremity of the maxilla.

As described above, each vomer bears a single tooth row containing four to eight teeth. The tooth row is essentially straight, but arranged obliquely medial to the choana with the two tooth rows slightly converging anteriorly. The tooth rows are set far apart from one another, corresponding to the separation of the vomers. Other palatal elements, including the parasphenoid and pterygoid, are entirely edentulous.

The dentary tooth row is slightly longer than that of the maxilla, terminating posteriorly at a point slightly anterior to the coronoid process of the prearticular. There are 20–25 teeth on the dentary. All marginal teeth and vomerine teeth are pedicellate, with the basal pedicel and crown separated by a poorly mineralized dividing zone. Tooth crowns are bicuspid as commonly seen in most salamanders.

Hyobranchial apparatus

The hyobranchial apparatus in B. londongensis has been shown to have two different patterns related to life-history differences (Fei et al., 2006: fig. 69). The neotenic pattern retains some of the larval branchial arches, and thus displays more complex structures than in the postmetamorphic pattern (Fig. 6; Fig. S3).

Figure 6 Hyobranchial apparatus of Batrachuperus londongensis (A) CIB 14499, showing a neotenic pattern with extra ceratobranchials III and IV; (B) CIB 14504, showing a postmetamorphic pattern with ceratobranchials III and IV entirely lost by resorption.

Cartilaginous elements cannot be observed in our CT-scanned images, but can be detected in cleared and stained specimens (Fig. 6). A cornua is present as a partly mineralized median plate in the postmetamorphic pattern, whereas it is absent in neotenes (Figs. 6A and 6B). In both neotenic and postmetamorphic individuals, the paired radial loops stemming from basibranchial I (anterior copula) extend anteriorly, then curve laterally in continuation with the ceratohyal. Therefore, the paired radial loops do not cross one another, in contrast to B. pinchonii and some other hynobiids (Onychodactylus, Hynobius, Pseudohynobius, Pachyhynobius), in which a Fig. 8-shaped pattern is formed by a complex crossing of the radial loops (Larson, Beneski & Miller, 1996; Rose, 2003; Xiong et al., 2013b).

Figure 7 Holotype skeleton of Batrachuperus londongensis (CIB 65I0013/14380) 3D reconstruction of whole body of the holotype skeleton in dorsal (A) and ventral (B) views.

Basibranchial I (anterior copula) remains cartilaginous in both neotenic and postmetamorphic individuals as in all other hynobiids except Onychodactylus, in which the element is partly ossified as a short stub (Fukuda, 1930; Smirnov & Vassilieva, 2002; Xiong et al., 2013b). Presence or absence of basibranchial II (posterior copula, os thyroideum of Rose, 2003) was previously unknown for B. londongensis. Observation of multiple specimens in this study reveals that basibranchial II is well ossified, displaying two different patterns that may reflect developmental differences (see below): most specimens have an anteriorly directed rod jointed with a transverse bar to form a simple inverted “T”-shaped structure (Fig. 6B; Figs. S3A–E); other specimens have a long anterior rod connected with a complex base, which displays multiple short branches directed posteriorly, termed here the “fork”-shaped pattern (Fig. S3F). None of the specimens on which this study is based shows an anchor-shaped basibranchial II, a configuration that has been recognized as a plesiomorphic condition for urodeles (Jia & Gao, 2016b). The anchor-shaped basibranchial II occurs in the stem caudate Karaurus (Ivachnenko, 1978; Estes, 1981), basal cryptobranchoid Chunerpeton (Gao & Shubin, 2003), and basal salamandroid Beiyanerpeton (Gao & Shubin, 2012). The plesiomorphic pattern is not seen in any extant hynobiids (see Xiong et al., 2013b), but occurs in the basal hynobiid Nuominerpeton (Jia & Gao, 2016a).

Hypobranchial I and ceratobranchial I remain separate in B. londongensis, in contrast to the fusion in B. pinchonii, B. tibetanus (Fei et al., 2006), and several other hynobiids (Ranodon, Hynobius, Pseudohynobius, Salamandrella, Liua) (Fei et al., 2006; Zhang et al., 2009; Xiong et al., 2013b). Hypobranchial I in most specimens remain cartilaginous, but ossification of the element occurs in large neotenes (CIB 65I0013/14380, 14381). CIB 14504 is the only postmetamorphic individual that has it as a bony rod (Fig. 6B; Fig. S3B). Ceratobranchial I is increasingly ossified ontogenetically in neotenes (CIB 65I0013/14380, 14381, 14482, 14484, 14485, 14499), but it remains cartilaginous in all postmetamorphic individuals. Whether bony or cartilaginous, the distal end of ceratobranchial I is slightly hooked, bearing a small process that curves posterodorsally (Figs. 6 and 7; Video S1).

Figure 8 Pectoral girdle and upper arm of Batrachuperus londongensis CIB 65I0013/14380, left scapulocoracoid in lateral (A) and lateroventral (B) views; left scapulocoracoid of CIB 14381 in lateral (C) and lateroventral (D) views; left scapulocoracoid of CIB 14504 in lateral (E) and lateroventral (F) views; left humerus of CIB 65I0013/14380 in dorsal (G) and ventral (H) views; right humerus of CIB 14381 in dorsal (I) and ventral (J) views; left humerus of CIB 14504 in dorsal (K), and ventral (L) views.

Note unusual ossification of normally cartilaginous parts of procoracoid and coracoid in the holotype (A, B) and CIB 14504 (E, F). All images not to scale.

Hypobranchial II and ceratobranchial II are both ossified as separate elements as in all other hynobiids (Fei et al., 2006; Xiong et al., 2013b). The former element is basically rod-like, slightly thicker posteriorly than anteriorly; the shaft can be relatively straight or slight curved laterally. Ceratobranchial II is much longer than hypobranchial II, and often displays an expanded distal end, which is bilaterally compressed into a distal plate. This is especially the case in large neotenes, including the holotype. At midlength in the long rod, a prominent process projects dorsomedially where the subarcualis rectus II attaches as in Amphiuma (Erdman & Cundall, 1984). Ceratobranchials III and IV are normally absent in postmetamorphic individuals (but see below), whereas they are present as bony elements in neotenes (Figs. 6 and 7). CIB 14487 is a postmetamorphic young adult (TL = 164 mm; SPL = 89.27 mm); its adulthood is indicated by ossification of the articular in the lower jaw, but it displays remnant ceratobranchial III in the process of being resorbed (Fig. S3E).

The ceratohyal in both postmetamorphic and neotenic individuals is distally ossified, with large adults showing more extensive ossification than smaller ones. In the holotype (CIB 65I0013/14380), the largest specimen known for the species (TL = 265 mm) and a neotene, over 60% of the ceratohyal is ossified as a robust element, leaving only the anterior one-third cartilaginous (Fig. 7B; Fig. S3A). The ossified part of element has a prominent ridge posterodorsally bearing a ligamentous connection with the suspensorium (Trueb, 1993; Rose, 2003). Partial ossification of the ceratohyal is also seen in all other species of Batrachuperus: B. pinchonii (Zhang et al., 2009: fig. 24); B. tibetanus (FMNH 5901); B. yenyuanensis (FMNH 49371); B. karlschmidti (FMNH 49380); B. taibaiensis (CIB 20040235). Among other hynobiids, partial ossification of the ceratohyal distally occurs in Pachyhynobius shangchengensis, Liua shihi, and Ranodon sibiricus (Zhang, 1985; Fei et al., 2006).

Axial skeleton

As consistently observed from all B. londongensis specimens studied here, the vertebral column consists of 18 presacral vertebrae, including the atlas, plus a single sacral, three to four caudosacrals, and 29–30 caudal vertebrae (Fig. 7; Video S2). The number of the presacral vertebrae is different from the type species, B. pinchonii (Zhang et al., 2009; Xiong et al., 2013a; but see Litvinchuk & Borkin, 2003: 16–17), as well as several other congeneric species (B. tibetanus, B. karlschmidti, and B. yenyuanensis), which have 17 presacrals by our own observation. Among other hynobiid species, the number of presacral vertebrae ranges from 15 in Ranodon sibiricus to 22 in Onychodactylus fischeri (Litvinchuk & Borkin, 2003).

In both dorsal and ventral views, the atlas is widened anteriorly, where it bears a pair of cotyles that receive the occipital condyles. Anteroventral to the cotyles is the odontoid process (tuberculum interglenoideum), which bilaterally articulates with the exoccipitals at the rim of the foramen magnum. Posterior to the odontoid process, the ventral surface of the atlas is penetrated by a pair of foramina through which the first pair of spinal nerves pass. The atlas has no transverse process or free ribs.

In all trunk vertebrae, the centrum is more or less cylindrical and is deeply amphicoelous. There is no significant change in length of the centrum along the trunk series, and all trunk vertebrae bilaterally bear transverse processes that are directed posterolaterally. A shallow groove is visible on the posterior surface of the transverse process, indicative of fusion of the dorsal diapophysis with the ventral parapophysis. Because of this fusion, the articulation facets of the diapophysis and parapophysis are confluent to receive the unicapitate ribs.

All presacral vertebrae except the atlas articulate with free ribs. The ribs associated with the first three trunk vertebrae are more robust than the remaining ribs, having the distal end expanded where the M. thoracic-scapularis attaches (Francis, 1934). In addition, the first pair of ribs often bears a short uncinate process distally, so that the ribs are distally forked (e.g., CIB 14482, 14484, 14485, 14487; Fig. S3; Video S2). The ribs are similar in length for most of the trunk series, but the posterior four to five pairs are abruptly shortened, with the last pair being a remnant stub that is even shorter than the transverse process of the associated vertebra. All trunk ribs are essentially single headed, although a shallow groove can be recognized posteriorly on the rib head as indication of fusion of the dorsal tuberculum with the ventral capitulum. Possession of unicapitate ribs is a diagnostic feature of the suborder Cryptobranchoidea (Dunn, 1923; Duellman & Trueb, 1986; Gao & Shubin, 2012; Jia & Gao, 2016a).

The single sacral vertebra is roughly the same size as the presacrals, but the transverse process of the sacral is obviously more robust than that of the trunk vertebrae. The sacral rib is elongated to at least twice the length of the transverse process, and curves ventrally to bear a ligamentous connection with the ilium.

The three rib-bearing vertebrae following the sacral are identified as caudosacrals, with the first haemal arch bearer marking the position of the anus. CIB 14485 is the only examined specimen that displays four caudosacrals. These caudosacrals are roughly the same size as the sacral vertebra, but ribs associated with them are greatly reduced, being slightly shorter than the corresponding transverse process. Possession of three or more caudosacrals seems to be plesiomorphic in urodeles (Gao & Shubin, 2001, 2012; Jia & Gao, 2016b). In fully grown adults, 28–30 vertebrae follow the caudosacral series (Fig. 7; Video S2). The caudal vertebrae normally have no free ribs, but occasionally remnant ribs occur in the first caudal in some specimens (CIB 14484, 14487). All caudal vertebrae, except the posterior-most two or three, ventrally bear a haemal arch, through which the haemal vessels pass.

Appendicular skeleton

Pectoral girdle and forelimbs

A large part of the pectoral girdle remains cartilaginous, with only the scapulocoracoid co-ossified as a single unit. A co-ossified scapulocoracoid occurs in all other salamanders but sirenids, in which the scapula and coracoid remain separate (Noble, 1931). The scapulocoracoid in B. londongensis has an extremely short scapular blade as in other species of the genus, but the dorsal border of the blade bears a distinct process projecting posterodorsally (Fig. 8; Video S2), a feature not seen in other Batrachuperus species. Because this process is consistently prominent in all specimens under study (Table 1), it is recognized here as a diagnostic feature of the species. Judging from its posterodorsal position, it unlikely served as insertion of the opercular muscle, but more likely as ligamentous insertion of a levator muscle of the shoulder; whereas it is the levator scapulae inferior or the intertransversaricus capitis inferior needs to be further investigated (see Gaupp, 1898; Kingsbury & Reed, 1909; Goodrich, 1930; Monath, 1965). Anterior to this process, the dorsal border of the scapular blade displays an elliptical depression that articulates the suprascapular cartilage.

In the ventral part of the pectoral girdle, the procoracoid and coracoid are fully co-ossified to form a large plate, which is expanded anteroposteriorly and curved ventromedially. In lateral view, the anterior border of the coracoid plate is straight, but the ventral border is rounded. Both the anterior and ventral borders are grooved to receive the cartilaginous part of the procoracoid and coracoid, respectively. Close to the posterodorsal border of the coracoid plate is a large glenoid fossa, which receives the head of the humerus. The glenoid fossa is roughly circular, but its anteroventral rim is deeply notched where the crista ventralis of the humerus fits (see below). Anteroventral to the glenoid fossa, a small supracoracoid foramen (occasionally two openings) penetrates the coracoid plate as passage of the supracoracoideus nerve (from the second and third spinal nerves) and its associated vessels (Francis, 1934). In addition, it is interesting to note that some specimens (e.g., CIB 65I0013/14380, 14504) display partial ossification of what otherwise is the cartilaginous part of the procoracoid and coracoid (Fig. 8; Video S2). Because extra ossification of these cartilaginous parts occurs only in very large specimens of both postmetamorphic and neotenic individuals, it can be viewed as a developmental feature but one that cannot be tied with neoteny or metamorphosis.

The humerus is significantly longer than the ulna or radius. Proximally on its extensor aspect, the crista dorsalis (dorsal crest) bears a prominent triangular or even knob-like projection, onto which the M. subscapularis inserts (Francis, 1934). Proximally on the flexor aspect, the crista ventralis (ventral crest) is a large triangular process, projecting anteroventrally where the M. pectoralis and M. supra-coracoideus attach (Francis, 1934); the ventral crest merges with the head of the humerus (or a facet in some specimens; see below). When the forelimb swings forward, the crista ventralis fits into the notch of the anteroventral rim of the glenoid fossa as mentioned above. Immediately ventral to the humeral head, a small but deep ventral depression marks the insertion of the flexor muscle M. humero-artibrachialis (Francis, 1934). In young adult specimens (e.g., CIB 14484, 14485, 14487), the humerus head is a cartilaginous cap, and thus, CT-scanned images show a shallow facet; in fully grown adults (e.g., CIB 65I0013/14380, 14381, 14482, 14504, 14509), the humeral head is ossified into a large spherical condyle (Figs. 7 and 8G–8L). Among the latter, no postmetamorphic specimens show a cartilaginous cap, whereas only relatively small neotenic individuals show a cartilaginous condyle. Distally, the olecranon fossa on the extensor side of the humerus is extremely shallow, barely recognizable, but the fossa cubitalis ventralis on the flexor side is a deep triangular depression, which receives the radius when the forearm is flexed. In relatively small neotenes, the radial and ulnar condyles are unossified, with the distal end of the humerus bearing two facets, where the cartilaginous ulnar and radial condyles attach.

In the forearm, both the ulna and radius are straight, subequal in length and approximately two-thirds the length of the humerus. The proximal end of the ulna is more expanded than its distal end, and conversely the radius is a more expanded distally. Posterolaterally, the ulna bears a bony crest that serves as insertion of the extensor of the forearm (M. extensor antibrachii ulnaris). The radius is distally in articulation with the radiale and intermedium. Close to its proximal end, the radius posterodorsally bears a prominent process, which serves as ligamentous insertion of the M. humero-antibrachialis, the main flexor muscle of the elbow (Francis, 1934). As described above for the humerus, in those specimens having a well-ossified humeral condyle, the epiphyses of the radius and ulna are also ossified, leaving no space for a cartilaginous cap.

Maximum ossification of the mesopodium in the forelimb displays a total of seven elements as observed in several specimens including the holotype (CIB 65I0013/14380, 14381, 14482, 14505, 14507, 14509), whereas other specimens (CIB 14484, 14485, 14487) have six ossified elements, with preaxial elements (radiale + element y) remaining cartilaginous (Figs. 9B and 9C). In those specimens showing maximum ossification, the intermedium proximally wedges between the ulna and radius, and distally wedges between the centrale and ulnare (Figs. 9A, 9B and 9D). The large radiale may represent a fusion of the radiale with element y, because no space for a possible element y is present in these specimens. The single large centrale is in direct contact with the radius, thus separating the radiale from contact with the intermedium. This pattern is also seen in all other congeneric species, including Batrachuperus yenyuanensis (FMNH 49371), which is the only known species of the genus in which two centralia occur. In B. londongensis, the radiale is slightly smaller than the centrale, and articulates with the radius proximally, the basale commune distally, and the centrale laterally. The basale commune is a large element, roughly triangular in shape. It is in articulation with the centrale proximally and with metacarpal 1 and 2 distally. Distal carpal 3 is slightly smaller than distal carpal 4, and the two elements articulate with metacarpal III and IV, respectively.

Figure 9 Left forearm of Batrachuperus londongensis (A) CIB 14381; (B) CIB 14484; (C) CIB 14487; (D) CIB 14507.

All images display presence of direct contact of the centrale with radius as an unusual feature in urodeles. All in dorsal view and not to scale.

There are four digits in the manus, the pattern seen in most salamanders. Digit 3 is the longest, having an extra phalange in comparison to the other digits. The phalangeal formula is 2-2-3-2, a generalized pattern for most salamanders (Shubin & Wake, 2003). The terminal phalanx of each digit is covered with a cornified sheath, possibly an adaptive feature correlated to living in a mountain stream environment.

Pelvic girdle and hind limbs

The pelvis in B. londongensis displays the usual pattern that occurs in most other salamanders: the paired ilia and ischia are ossified, whereas the pubis remains cartilaginous (but see below). In addition, the ypsiloid cartilage is present as in some but not all salamanders (see Hecht & Edwards, 1976, 1977). The ilium is roughly club-shaped, with a slightly thickened and expanded ventral plate and a narrow dorsal blade. The ventral plate contributes to a large part of the acetabulum and is ventrally in articulation with the ischium. The iliac blade is connected with the sacral rib by a ligament. The bone is set in a more or less vertical position, whereas the iliac blade is slightly inclined posteriorly.

The ischium is a large bony plate that is the main ventral element in the pelvis. The two ischial plates meet ventrally to form a symphysal hinge articulation along the midline. In ventral view, the ischium is anteriorly widened with a rounded border, to which the cartilaginous pubis is attached (but see below), but projects laterally at the posteroventral border of the acetabulum. Immediately posterior to the acetabulum, the lateral border of the plate is deeply notched between the acetabulum and a large posterolateral process (ischial spine). The latter process serves as origin of the flexor of the tail, M. ischio-caudalis (Francis, 1934). The elongate ischial spine is significantly different from that in the type species B. pinchonii, in which the spine is rudimentary (see Zhang et al., 2009).

The pubis often remains cartilaginous, as observed in several specimens (CIB 14482, 14484, 14485, 14487, 14507); however, in other specimens (CIB 65I0013/14380, 14381, 14504, 14509), the pubis is partly ossified, evidenced by the presence of a robust process anteroventral to the acetabulum (Figs. 10A–10D; Fig. S5). This interpretation is supported by the presence of an obturator foramen at the base of the process, which marks the border between the pubis and ischium (Fig. 10B). In addition, two of these specimens (CIB 14381, 14509) show that the fusion of the ossified part of pubis to the ischium is not yet completed; thus, in these two specimens the obturator foramen is still partly open anteromedially (Fig. 10D). In them, the large ventral plate in the pelvis actually represents the ossification of the ischium and at least a part of the pubis, and can best be termed a pubio-ischium.

Figure 10 Pelvis and femur of Batrachuperus londongensis Pelvis of CIB 65I0013/14380 in right lateral (A) and ventral (B) views; pelvis of CIB 14381 in right lateral (C) and ventral (D) views; pelvis of CIB 14482 in right lateral (E) and ventral (F) views; pelvis of CIB 14487 in right lateral (G) and ventral (H) views; left femur of CIB 65I0013/14380 in dorsal (I) and ventral (J) views; left femur of CIB 14381 in dorsal (K) and ventral (L)views; left femur of CIB 14482 in dorsal (M) and ventral (N) views; left femur of CIB 14487 in dorsal (O) and ventral (P) views.

All images not to scale.

It is also interesting to note that many specimens including the holotype display a second pair of small foramina more posteriorly located at the base of the ischial spine (Figs. 10B, 10D, 10F and 10H). To our knowledge, there are no publications that figured or described this pair of foramina, but they are indeed present in both neotenic and postmetamorphic individuals of B. londongensis.

A Y-shaped cartilage (ypsiloid) is not shown in CT-scanned images, but is observed in cleared and stained specimens (Fig. S5). This cartilage has been shown to be correlated with the hydro-static function of the lungs (Whipple, 1906); that is, the salamander can easily raise its head in the water by elevating the ypsiloid cartilage, thereby compressing the lower abdomen and forcing the air in the lungs forwards to the anterior end; conversely, the salamander is able to depress its head by lowering the ypsiloid cartilage, returning the air to the posterior portion of the lungs, enabling the animal to swim to deeper water. The ypsiloid cartilage is present in Hynobiidae, Cryptobranchidae, Salamandridae, and Ambystomatidae, but absent in Proteidae, Plethodontidae, Sirenidae, and Amphiumidae (Hecht & Edwards, 1976, 1977).

The femur is a robust element roughly the same length as the humerus. As seen in the holotype and several other specimens (CIB 65I0013/14380, 14381, 14504, 14507, 14509), the strongly expanded proximal end bears a large bony condyle in articulation with the acetabulum (Figs. 10I–10L). In four other specimens (CIB 14482, 14484, 14485, 14487), however, the femur is capped with cartilage; thus, CT images of these specimens display a gap between the femur and acetabulum. On the extensor side, the femur bears a longitudinal ridge, extending from the proximal end to the fibular condyle; this ridge serves as insertion of the M. pubo-ischio-femoralis internus, a powerful extensor of the thigh (Francis, 1934). Posterior to this ridge and close to the proximal end, a small tubercle marks the external trochanter (Figs. 10I and 10O), onto which the M. caudalis-femoralis inserts (Francis, 1934). On the flexor side, the robust femoral trochanter projects ventromedially as a twig-like process (Fig. 10I–10P), onto which the M. pubo-ischio-femoralis externus attaches (Ashley-Ross, 1992). The trochanteric crest extends from the femoral trochanter distally along the shaft to the tibial condyle. On the distal end, both the tibial and fibular condyles are well ossified to display a bony articulation with their corresponding elements. The tibial condyle is much larger than the fibular condyle. As described above for the proximal head, in some specimens (CIB 14482, 14484, 14485, 14487) the tibial and fibular condyles are poorly ossified, with slightly concave depressions as indicative of a cartilage-capped epiphysis.

The tibia and fibula are similar in length, but the former element is the more robust (Fig. 11). Proximally on its extensor aspect, the tibia bears a weakly developed tibial crest that serves as a tendinous insertion of the M. extensor ilio-tibialis (Francis, 1934). Along the distal half of the tibia is a well-defined bony ridge facing the fibula; this medial ridge probably serves as insertion of the M. extensor cruris tibialis (Ashley-Ross, 1992). At the proximal end of the medial ridge, a small but twig-like process provides a tendinous insertion of the M. pubo-ischio-tibialis (Francis, 1934). The lateral side of the fibula is straight but the medial side concave, with its distal end expanded medially to articulate with the tibia.

Figure 11 Lower hind limb of Batrachuperus londongensis (A) left lower hind limb of CIB 65I0013/14380; (B) left lower hind limb of CIB 14381; (C) left lower hind limb of CIB 14482; (D) left lower hind limb of CIB 14487.

All in dorsal view and not to scale.

Maximum ossification of the mesopodium includes as many as nine tarsal elements (Fig. 11). As observed in several specimens (CIB 65I0013/14380, 14381, 14485, 14509), the intermedium is a large element with its proximal process wedged between the tibia and fibula; the intermedium distally articulates with a single centrale. The fibulare is roughly the same size as the intermedium and is medially in extensive articulation with the latter. The tibiale is much smaller than the fibulare, but medially articulates with the centrale. Among distal elements, element y is more or less rounded, in articulation with the tibiale proximally, and with a large basale commune laterally. Distal tarsal 3 and 4 are similar in size and both are in contact with the centrale. Distal tarsal 5 is missing, what is likely related to the loss of digit 5. Distolaterally, the postminimus is a small bone in contact with both the fibulare and metacarpal IV. According to Shubin & Wake (2003), a postminimus is present in Batrachuperus and several other hynobiids (Liua, Ranodon, Salamandrella, Paradactylodon), and also in the cryptobranchid Andrias.

Several other specimens show less extensive ossification in the mesopodium than those described above. As observed in four specimens (CIB 14482, 14482, 14484, 14487), all those elements ossified already show essentially the same arrangement pattern as described above, but have one or two of the preaxial elements (tibiale and element y) still unossified. In addition, in none of these specimens is the postminimus ossified. Comparison of these specimens with those described above indicates that element y ossifies before the tibiale, whereas the postminimus is ossified after the tibiale; thus, the postminimus is the last tarsal element to be ossified ontogenetically.

As in all congeneric species, B. londongensis displays a reduction of hind limb digits from five to four, with digit 5 missing. The phalangeal formula is 2-2-3-2, except for two specimens (CIB 14484, 14504) that show developmental abnormalities (see below). The terminal phalanges are covered with a cornified sheath as occurs commonly in other mountain stream salamanders, including the Middle Eastern stream salamander Paradactylodon (Kami, 1999).

As mentioned above, two specimens (CIB 14484, 14504) show developmental abnormalities in the hind limb. In CIB 14484, the right foot has four digits, but five in the left. In keeping with the presence of digit 5, a distal tarsal 5 is also present in the left foot but not in the right. The digital formula in this specimen is 2-2-2-2-2 in the right foot, and 2-2-3-2 in the left. The other specimen (CIB 14504) has only three digits in left foot, with digit 1 entirely missing; but there are four digits in the right. The left foot has only six tarsal elements, whereas the right has 10, including two centralia. The six tarsals in the left may reflect abnormal fusion of several bones: possibly fusion of the intermedium with the centrale, and fusion of distal tarsals 2 + 3. The fibulare is enlarged and roughly the same size as the intermedium. Both left and right limbs have a small postminimus, indicative of this individual being a fully grown adult. The digital formula of this specimen is ?-2-3-2 for the left foot, and is 2-2-2-2 for the right.

Discussion

The Longdong Stream Salamander, B. londongensis, is a rare hynobiid species that is of special interest because of a life-history that features facultative neoteny. In addition, undertaking an osteological study of this salamander is urgent, because of its rarity and current vulnerability to extinction (Jiang et al., 2016). From comparison with other species of the genus and other hynobiids, our study has revealed several osteological features that are diagnostic of B. londongensis, while several other features are shown to be of developmental, ecological, or phylogenetic significance. We provide a discussion of these results below.

Osteological characterization of B. londongensis

Previously, B. londongensis had been diagnosed with reference only to external morphological characters and the number of chromosomes (Liu & Tian, 1978; Fei, Ye & Tian, 1983; Fei et al., 2006). In supplementing the diagnosis of this species by inclusion of osteological features, our study shows that some of these diagnostic features are also phylogenetically significant in respect to both hynobiids and to the evolution of salamanders more generally. Osteologically, B. londongensis shares with its congeneric species the following derived characters: the maxilla is extremely short, posteriorly terminating at a level anterior to midlevel of the orbit; the lacrimal extends anteriorly over the border of the external naris, but not over the orbital border posteriorly (except for B. taibaiensis); a nasal-maxillary contact is absent because the two elements are separated by the lacrimal; the otic process of the pterygoid has no bony contact with the parasphenoid (except for B. yenyuanensis); the ceratohyal is distally ossified; a single centrale (except for B. yenyuanensis) in the manus is in direct contact with the radius, preventing the radiale from contacting the intermedium; the number of digits is reduced from five to four in the pes (shared with Salamandrella and Paradactylodon as homoplasies); terminal phalanges are covered with a cornified sheath (shared with Onychodactylus, Liua, Paradactylodon as ecological homoplasies).

Batrachuperus londongensis is distinguished from B. pinchonii and all other congeneric species by a combination of the following characters: alary process of premaxilla barely contributing to border of anterodorsal fenestra (shared with B. tibetanus); suture between nasal and frontal located at the level of anterior border of orbit; vomers do not contact at the midline (unique); vomerine teeth four to eight in number, arranged in a straight line that is nearly vertically oriented; the ascending process of the palatoquadrate is ossified as a pillar between the parietal and the otic process of the pterygoid (shared with B. tibetanus and B. taibaiensis); radial loops stemming from basibranchial I do not cross one another (shared with B. pinchonii, B. tibetanus; unknown for other species); cartilaginous hypobranchial I and ceratobranchial I remain separate (fused in B. pinchonii and B. tibetanus); presacral vertebrae 18 in number; the scapular blade is extremely short, with a prominent posterodorsal process (unique); the ischial plate is penetrated by a small nerve foramen, and the ischial spine is clearly more elongated than in other species.

Developmental features related to life-history differences

As stated above (see Introduction), B. londongensis may represent the only living hynobiid that is facultatively neotenic, as both biphasic individuals and neotenes (“permanent larval producers” of Rose, 2003) are known for the species. Based on our study of both neotenic postmetamorphic specimens, we provide a discussion of life-history differences of the species as presented below.

In terms of body size, neotenes tend to be larger than postmetamorphic individuals. The holotype CIB 65I0013/14380 (TL = 265 mm) is the largest individual known for the species, but still retains gill slits and a larval configuration of the hyobranchium (Fei et al., 2006). Because many relatively small specimens (TL = 162–241 mm) have completed metamorphosis, the presence of external gill slits and retention of a larval hyobranchium in large individuals cannot be logically interpreted as ontogenetic features always leading to metamorphosis in B. londongensis. By contrast, the holotype and several other large individuals display fully ossified limbs and a postminimus in the pes, thereby giving a clear indication that these are fully grown adults, in spite of their retention of some larval features.

The most striking morphological difference between neotenic and postmetamorphic individuals is seen in the hyobranchium. As figured in Fei et al. (2006: fig. 69f, g), neotenes always display a more complex hyobranchium than do postmetamorphic individuals, in which ceratobranchials III and IV are entirely missing. Among median elements, the inverted “T-shaped” basibranchial II occurs in both postmetamorphic and neotenic forms, whereas the “fork”-shaped pattern is seen only in some neotenes as a developmental feature. We provide a possible interpretation here: the “fork”-shaped pattern can be viewed as the initial larval pattern, with the posterior branches subsequently absorbed at metamorphosis, whereas in neotenes the absorption process is prolonged or delayed. This interpretation is supported by evidence that large neotenic individuals, including the holotype CIB 65I0013/14380, which display the inverted “T”-shaped pattern, still have a remnant of the posterior processes, whereas slightly smaller and presumably younger adults show the “fork”-shaped pattern.

With respect to the laterally paired elements, both neotenic and postmetamorphic individuals show incomplete ossification of the ceratohyal, with large neotenes displaying more extensive ossification than smaller ones. This observation indicates that ossification of the ceratohyal terminates at metamorphosis, but continues in neotenes. Hypobranchial I is partly ossified in large neotenes, but the element is missing in typical postmetamorphic forms (see below). Ceratobranchial I is partly ossified only in neotenes. Evidently, ossification of the first branchial arch derivatives is terminated at metamorphosis, but is prolonged in neotenes. Ossification of hypobranchial II and ceratobranchial II is a common pattern in all hynobiids, whereas ceratobranchials III and IV are retained only in neotenic individuals, and are normally absent in postmetamorphic forms. CIB 14487 is a relatively small postmetamorphic adult, but it retains a remnant part of the ceratobranchial III bilaterally. Comparison of this specimen with those that show the typical postmetamorphic pattern of the hyobranchium indicates that resorption of larval arches can be prolonged until after metamorphosis is otherwise completed.

Partial ossification of ceratohyal in hynobiids

Comparison of B. londongensis with congeneric species and other hynobiids suggests that partial ossification of the ceratohyal is a feature of phylogenetic significance. Within Batrachuperus, a similar pattern of ossification of the ceratohyal occurs in B. tibetanus (FMNH 5901), B. pinchonii (Zhang et al., 2009: fig. 24), B. karlschmidti (FMNH 49380), B. yenyuanensis (FMNH 49371), and B. taibaiensis (CIB 20040235). Xiong et al. (2013b: fig. 1E and 1H) illustrated B. pinchonii and Liua shihi as having the ceratohyal entirely cartilaginous, but characterized this structure in the description as partly ossified in both species. Based on our own observations, we conclude that partial ossification of the ceratohyal occurs in all species of Batrachuperus.

In other hynobiids, partial ossification of the ceratohyal occurs in the monotypic Pachyhynobius shangchengensis and in Ranodon sibiricus (Fei & Ye, 1983; Fei et al., 2006; however, Xiong et al., 2013b: fig. 1G shows no ossification in R. sibiricus), and in Liua shihi (Zhang, 1985; Fei et al., 2006; contra Xiong et al., 2013b: fig. 1, but mentioned as partly ossified in the description). In addition, our observation of a specimen of Paradactylodon mustersi (FMNH 211936) shows that it also displays a partly ossified ceratohyal. Another species, Paradactylodon persicus (MVZ 241494; AmphibiaTree, 2004) shows no ossification of the ceratohyal, but the specimen seems to be a young adult, as the nasals do not yet fully meet at the midline and a small parietal fontanelle is not yet completely closed. Ossification of the ceratohyal in four hynobiid genera (Onychodactylus, Salamandrella, Pseudohynobius, Hynobius) has yet to be recorded.

In the closely related clade Cryptobranchidae, the distal end of the ceratohyal is ossified in the North American hellbender Cryptobranchus alleganiensis (Elwood & Cundall, 1994: fig. 9; Rose, 2003: fig. 5), but not in the Chinese or Japanese giant salamander Andrias (Sato, 1943; Aoyama, 1930; Wu, 1982; Rose, 2003). Among non-cryptobranchoid salamanders, partial ossification of the ceratohyal is seen in some salamandrids (e.g., Notophthalmus viridescens) and some dicamptodontids (e.g., Dicamptodon ensatus), in Siren and Pseudobranchus, and in Proteus but not in Necturus (see Rose, 2003 for citations). Clearly, the distribution of this character needs to be scrutinized in a cladistic analysis to understand its phylogenetic significance.

Ossification patterns of mesopodial elements and reduction of number of digits

Salamander limbs display variant structural patterns that provide significant information for addressing questions of morphological evolution in terms of ecological adaptation (Shubin & Wake, 2003). In this context, several limb features of B. londongensis, including the arrangement of mesopodial elements and the reduction of the number of digits, are worth consideration in comparison with other congeneric species and other hynobiids as well.

Patterns of limb structure in hynobiid salamanders have been documented for several genera (Onychodactylus, Batrachuperus, Salamandrella, Liua, Ranodon, Paradactylodon), but remain largely unknown or poorly known for several others (Hynobius, Pachyhynobius, Pseudohynobius). Batrachuperus has been regarded as having two centralia (Shubin & Wake, 2003), but this interpretation was probably based on B. yenyuanensis, which indeed has two centralia; however, this study revealed that all other species of the genus, including the type species B. pinchonii, have a single centrale in both manus and pes. Although the single centrale is in direct contact with the radius in the forelimb, it does not contact the tibia in the hind limb. A direct contact of the centrale with the radius has been recognized as a peculiar feature in extant salamanders, as it also occurs in some lepospondyls and temnospondyls (Carroll, 1968; Carroll & Gaskill, 1978; Shubin & Wake, 2003). Regardless, B. yenyuanensis is the only species within Batrachuperus that possesses two centralia. Phylogenetic analysis based on molecular data has shown that B. yenyuanensis occupies a basal position in relation to other species of the genus (Chen et al., 2015); thus, the single centrale in these other species is likely to be a derived state acquired within Batrachuperus. Among other hynobiids, retention of two centralia is known for several taxa, including Liua, Ranodon, Salamandrella, and Paradactylodon (Shubin & Wake, 2003).

Comparison of specimens of Batrachuperus species reveals a delayed ossification of preaxial elements in both fore- and hind limbs. In the forelimb, fully grown adults of B. londongensis (e.g., holotype CIB 65I0013/14380, 14381, 14482, 14504) have a large radiale, which probably reflects fusion of the radiale with element y, whereas young adults (e.g., CIB 14484, 14485, 14487) show no ossification of the radiale or element y. Whether the radiale and element y are co-ossified or fused together in fully grown adults needs to be verified by developmental evidence. In the hind limb, fully grown adults (e.g., CIB 65I0013/14380, 14381, 14504, 14507, 14509) clearly show that the tibiale and element y are ossified as separate elements, with several young adults (e.g., CIB 14482, 14484, 14485) showing that element y is ossified before the tibiale. Comparisons with other species of Batrachuperus and other hynobiids are needed to understand the ontogenetic and phylogenetic significance of these patterns.

According to Shubin & Wake (2003), a postminimus is present in Batrachuperus, several other genera of hynobiid salamanders (Onychodactylus, Liua, Ranodon, Salamandrella, Paradactylodon), and also the cryptobranchid genus Andrias. Comparison of specimens of congeneric species in this study leads us to concur with Shubin & Wake (2003) in that congeneric species in Batrachuperus indeed have a postminimus in pes, and this is probably the last limb element to be ossified ontogenetically, as the postminimus is observed in fully grown adults but not in those young adults that have element y, or element y and tibiale ossified (see above). The postminimus is absent in Hynobius, Pachyhynobius, and Pseudohynobius (Shubin & Wake, 2003). Although relevant information on presence/absence of this element is still sketchy for some hynobiid taxa, we interpret that possession of a postminimus as a plesiomorphic feature because it is widely distributed in the family Hynobiidae. However, how many taxa at generic level share the loss of the element as a homology, or independent loss of this element need to be scrutinized in a thorough phylogenetic analysis.

Finally, B. londongensis displays limb patterns commonly seen in other mountain stream salamanders: The number of digits is reduced from five to four in hind limb, and terminal phalanges in both fingers and toes are covered with a cornified sheath. These features have been recognized as corresponding to ecological adaptations of the salamanders to higher altitude mountain stream environments (Fei & Ye, 1984). These limb features are commonly seen in all other Batrachuperus species, and in other hynobiids (Salamandrella, Paradactylodon, and some species of Hynobius), which are also mountain stream dwellers (Reilly, 1983; Fei et al., 2006). Interestingly, there are some hynobiids (Onychodactylus, Ranodon, Liua) that are mountain stream dwellers, but none of these show reduction of digits from five to four in the pes; and Onychodactylus is the only genus other than those mentioned above has developed claw-like terminal phalanges with a cornified sheath (Fei et al., 2006).

Conclusion

Our study of the osteological anatomy of B. londongensis has led to the following conclusions: Batrachuperus londongensis is diagnosed by a set of osteological features, including unique features: alary process of premaxilla is excluded from the border of the anterodorsal fenestra; vomers lack medial contact; parasphenoid enters anteromedial fenestra; vomerine tooth row nearly vertical in position; presacral vertebrae 18 in number; scapular blade develops a distinct posterodorsal process.

Batrachuperus londongensis is among the few hynobiids that display perichondral ossification of the ascending process of the palatoquadrate as a part of the suspensorium.

A stapedial foramen is unexpectedly present in some but not all specimens, an unusual feature that needs to be thoroughly investigated.

Neotenic individuals display a more complex structural pattern of the hyobranchium than postmetamorphic individuals, most notably the retention of ceratobranchials III and IV. Neotenes also show increased ossification of hyobranchial elements during aging, in contrast to loss of elements by resorption in metamorphic individuals.

Batrachuperus londongensis has a single centrale in direct contact with the radius in the manus, but not with the corresponding element, the tibia, in the pes. In both fore- and hind limbs, delayed ossification of preaxial elements is a common pattern in both neotenes and postmetamorphic specimens.

Batrachuperus londongensis retains element y and the postminimus in the pes as a plesiomorphic pattern in hynobiids. Phylogenetic significance of the retention or loss of these limb elements within Hynobiidae requires a thorough investigation.

Supplemental Information

Supplemental Information 1 Lateral view of the skull of Batrachuperus londongensis with squamosal and quadrate attached.

(A) CIB 65I0013/14380; (B) CIB 14381; (C) CIB 14482. Abbreviations used in this and other supplementary figures see Materials & Methods in main text.

Click here for additional data file.

Supplemental Information 2 Occipital of the skull of Batrachuperus londongensis with squamosal and quadrate attached.

(A) CIB 65I0013/14380; (B) CIB 14381; (C) CIB 14482.

Click here for additional data file.

Supplemental Information 3 Skull and mandibles of Batrachuperus londongensis in ventral view, with hyobranchium attached.

(A) CIB 65I0013/14380; (B) CIB 14504; (C) CIB 14484; (D) CIB 14507; (E) CIB 14487; (F) CIB 14482. Arrow in (A) and (E) pointing to the prominent process projecting from the midlength of the ceratobranchial II for attachment of the subarcualis rectus II muscle.

Click here for additional data file.

Supplemental Information 4 Vomers of Batrachuperus londongensis in palatal view:.

(A) CIB 65I0013/14380; (B) CIB 14381. Foramina labeled in images are for passage of the ramus ventralis of the trigeminal nerve (CN V) and the ramus palatinus of the facial nerve (CN VII) as described in main text.

Click here for additional data file.

Supplemental Information 5 Cleared and stained specimen (CIB 14499) showing the pelvic girdle of Batrachuperus londongensis in dorsal (A) and ventral (B) views.

Click here for additional data file.

Supplemental Information 6 Specimens used for comparison in this study.

Click here for additional data file.

Supplemental Information 7 CT rendered holotype skull of Batrachuperus londongensis (CIB 65I0013/14380).

Click here for additional data file.

Supplemental Information 8 Holotype whole skeleton of Batrachuperus londongensis (CIB 65I0013/14380).

Click here for additional data file.

We thank Yuezhao Wang (Chengdu Institute of Biology, Chinese Academy of Sciences, Chengdu, China), Alan Resetar (Field Museum of Natural History) for access to specimens under their care, and Zhe-Xi Luo (University of Chicago, Chicago, IL, USA) for use of the CT scanner in his laboratory. Professor RC Fox (University of Alberta, Edmonton, Canada) read and commented on the manuscript, and Yu-long Li (Chengdu Institute of Biology, Chinese Academy of Sciences, Chengdu, China) helped with CT scanning of some specimens. We also thank Neil Shubin (University of Chicago) for discussion on ossification patterns and evolution of the mesopodium in salamanders. Two reviewers (M. Laurin and J. Ziermann) provided constructive comments and suggestions that lead to improvement of our manuscript.

Institutional Abbreviations

CIB Chengdu Institute of Biology, Chinese Academy of Sciences, Chengdu, China

FMNH Field Museum of Natural History, Chicago, IL, USA

MVZ Museum of Vertebrate Zoology, University of California, Berkeley, CA, USA

Anatomical Abbreviations

act acetabulum

adf anterodorsal fenestra

amf anteromedial fenestra

an angular

ar articular

at atlas

bb basibranchial

bc basale commune

c centrale

cb ceratobranchial

ch ceratohyal

cor coracoid

corn cornua

crd crista dorsalis

crv crista ventralis

dc distal carpal

den dentary

dt distal tarsal

etr external trochanter

fe femur

ftr femoral trochanter

fi fibula

fib fibulare

fr frontal

glf glenoid fossa

hb hypobranchial

hu humerus

i intermedium

icf internal carotid foramen

il ilium

isc ischium

lac lacrimal

lacf lacrimal foramen

mx maxilla

na nasal

obf obturator foramen

obs orbitosphenoid

op-ex opisthotic–exoccipital complex

pa parietal

pcor procoracoid

pm premaxilla

po postminimus

pra prearticular

prf prefrontal

pro prootic

ps parasphenoid

pt pterygoid

pub pubis

qu quadrate

ra radius

rad radiale

rl radial loop

sca scapulocoracoid

scof supracoracoid foramen

sm septomaxilla

sq squamosal

st stapes

stf stapedial foramen

ti tibia

tib tibiale

ul ulna

uln ulnare

vo vomer

y element y

yps ypsiloid

Additional Information and Declarations

Competing Interests

Author Contributions

Data Availability

The authors declare that they have no competing interests.

Jian-ping Jiang conceived and designed the experiments, analyzed the data, contributed reagents/materials/analysis tools, authored or reviewed drafts of the paper, approved the final draft.

Jia Jia conceived and designed the experiments, performed the experiments, analyzed the data, contributed reagents/materials/analysis tools, prepared figures and/or tables, authored or reviewed drafts of the paper, approved the final draft.

Meihua Zhang performed the experiments, analyzed the data, contributed reagents/materials/analysis tools, prepared figures and/or tables, authored or reviewed drafts of the paper, approved the final draft.

Ke-Qin Gao conceived and designed the experiments, analyzed the data, prepared figures and/or tables, authored or reviewed drafts of the paper, approved the final draft.

The following information was supplied regarding data availability:

All specimens used in this study are stored in the collection at Chengdu Institute of Biology (CIB), Chinese Academy of Sciences.

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
