# Peer review of "Osteology of Batrachuperus londongensis (Urodela, Hynobiidae): study of bony anatomy of a facultatively neotenic salamander from Mount Emei, Sichuan Province, China"

_PeerJ, doi:10.7717/peerj.4517_

## Round 0.1 · original submission · Major Revisions

I have obtained reviews from two reviewers as attached below. Both of the reviewers find merit and novel results in this manuscript, but there are a number of issues (but not critical) to be addressed before publication. Please address all the points raised in the report as all are constructive and should improve the manuscript. I agree with reviewer 1 that the manuscript can be much improved by deepening the discussion. In particular, the evolutionary implications of the anatomical data that you provide. Please attempt to broaden and deepen the discussion, given the guidance provided by the reviewer. Reviewer 1 has also provided a separate annotated file, so please do not forget to incorporate these into the revision.

·

Basic reporting

The text is fairly clear and generally well-written. On the annotated pdf file (attached to this report), I have suggested several minor stylistic improvements, clarifications, and grammatical or typographic corrections.

The description is competent. On this front, I only regret that the authors did not provide more illustrations, especially that on-line journals like PeerJ have less stringent space limitations than their paper-based forebears. In this respect, I would very much like to see an occipital view of the skull, and if possible, a lateral view with the quadrate and squamosal in place. The ypsiloid cartilage, the zeugopod elements (radius, ulna, tibia, fibula) should be illustrated too.

I find the discussion of the paper weaker, and will concentrate the review on this aspect.
The authors do not provide a thorough overview of the evolutionary implications of the anatomical data that they provide. For instance, the seldom-reported ligament connecting the stapes to the quadrate is intriguing. Do taxa that have lost the tympanum (several squamates, some frogs) have this? If not, it would constitute some support for the lepospondyl hypothesis of lissamphibian origins (Laurin, 1998; Marjanovic & Laurin, 2013), because the temnospondyl hypothesis suggests that some distant urodele ancestors had a tympanum (Lombard & Bolt, 1979).
Similarly, the claim that the authors make that the absence of the septomaxilla is not paedomorphic is rather weak, to the extent that this bone is one of the last to ossify, and this appears to have been ancestrally present in urodeles (Germain & Laurin, 2009). Not all paedomorphic characters need to be present in all taxa; indeed, theirs lacks external gills…
The authors have some evidence about carpus and tarsus development, which bears on the pre- or post-axial dominance being ancestral to urodeles (Fröbisch & Shubin, 2011), but they do not discuss this issue; they should. Also, they apparently present only part of the evidence that they have, judging from line 1101, in the discussion. This is too bad; they should present it all and discuss this point more thoroughly.
In the introduction, the authors discuss the fossil record and molecular dates for various cryptobranchoid taxa, but they confuse two (divergence hynobiids/cryptobranchids, and evolutionary radiation of crown-hynobiids), so that the discrepancy between various estimates is even greater than they suggest. This needs to be clarified. Also, I noticed some discrepancies with the latest paleontological timetree for urodeles (Marjanovic & Laurin, 2014); it would be good to know why. Perhaps some more recent research refuted some of the taxonomic positions or stratigraphic age assignments suggested by Marjanovic & Laurin (2014)? The authors, who know these taxa well, could take this opportunity to provide an update.
The nomenclatural review (lines 235-249) leaves some issues dangling because it focuses on homonymy, but it looks like there are also synonymy issues, which are currently not discussed, but should be.
In the discussion, the presence of a contact between radius and centrale is mentioned in some temnospondyls, but the authors are apparently unaware that this also occurs in at least some lepospondyls, such as Pantylus (Carroll, 1968: fig. 2; Carroll & Gaskill, 1978: fig. 125). There is already an extraordinary bias towards the temnospondyl hypothesis in the literature; no need to enhance this by reporting only part of the evidence.

References
Carroll, R. L. (1968). The postcranial skeleton of the Permian microsaur Pantylus. Canadian Journal of Zoology 46, 1175-1192.
Carroll, R. L., and Gaskill, P. (1978) The order Microsauria. Philadelphia: American Philosophical Society. Fröbisch, N. B., and Shubin, N. H. (2011). Salamander limb development: integrating genes, morphology, and fossils. Developmental dynamics 240, 1087–1099.
Germain, D., and Laurin, M. (2009). Evolution of ossification sequences in salamanders and urodele origins assessed through event-pairing and new methods. Evolution & Development 11, 170–190.
Laurin, M. (1998). The importance of global parsimony and historical bias in understanding tetrapod evolution. Part I. Systematics, middle ear evolution, and jaw suspension. Annales des Sciences Naturelles, Zoologie, Paris, 13e Série 19, 1-42.
Lombard, R. E., and Bolt, J. R. (1979). Evolution of the tetrapod ear: an analysis and reinterpretation. Biological Journal of the Linnean Society 11, 19-76.
Marjanović, D., and Laurin, M. (2013). The origin(s) of extant amphibians: a review with emphasis on the “lepospondyl hypothesis”. Geodiversitas 35, 207–272.

Experimental design

Paleontology is not an experimental science, like the rest of morphology and systematics. It is a historical and descriptive science. Thus, most of the points mentioned in the form do not apply.

Validity of the findings

Findings seem valid. My only comment is that broader relevance the findings needs to be highlighted better by more thorough discussions of the most interesting findings. The current version is clearly under-developed, on this front. I made various comments in the section “Basic reporting” about this. I think that if the authors follow these suggestions, the paper will be very interesting.

Additional comments

Good work. I hope that you will find my comments useful to make it even better, and more importantly, to better highlight its relevance to a much broader community. Do not hesitate to contact me if you have questions.
Michel Laurin
michel.laurin@mnhn.fr

·

Basic reporting

Overall the manuscript is well written, however many minor changes are required as listed below. The Results are Results +Discussion and the Discussion is actually a summary. All figures are relevant but not sufficiently cited in the Results part.

Experimental design

The research / manuscript is relevant, and the authors considered all relevant literature. A list with all investigated specimens including also non- B. londongensis specimens would be helpful (supplementary table?).

Validity of the findings

The data can be replicated, and all data seem valid. Conclusions are clearly stated.

Additional comments

Basic reporting
Overall the manuscript is well written, however many minor changes are required as listed below. The Results are Results + Discussion and the Discussion is actually a Summary. All figures are relevant but not sufficiently cited in the Results part.

Experimental design
The research / manuscript is relevant, and the authors considered all relevant literature. A list with all investigated specimens including also non- B. londongensis specimens would be helpful (supplementary table?).

Validity of the findings
The data can be replicated, and all data seem valid. Conclusions are clearly stated.

List of suggested changes: GENERALL
- get rid of double-spaces between sentences
- Results: The paper has great figures, but the description (results) fails to refer to them  add Figures throughout Results where appropriate.
o It would be also easier to read/understand when you make a general anatomical description and then finish the results with a neotenic description where you only highlight what is different between postmetamorphic and neotenic specimens
o your results are mix of results and discussion and your discussion is a summary – consider changing the headings accordingly
o you have female and male specimens – are there gender differences? You only cleared and stained two females.
- Material & Methods:
o Add a supplementary table where you list the specimens you compared with your CIB samples
o line 135-136 add numbers for the specimens: e.g., juvenile (1?), adults (11?), postmetamorphic (4), neotenic (8)
o Introduce abbreviation FMNH as you did with CIB (line 141)
- change Batrachuperus londongensis to B. londongensis in following lines: 23, 41, 103, 123, 124, 128, 153, 217, 246, 247, 264, 355, 678, 729, 976, 1013, 1136
- refer to abbreviation list at and of your figure legends Fig. 2 to Fig. 11
- add spaces after CIB: line 1476, Table 1
- check if you mean protractor pectoralis in lines 32, 227, 782
- throughout text change phrasings like “structure FOR attachment (origin, insertion) of muscle xy” to “structure, onto which muscle xy attaches/inserts” or to “structure, from which muscle xy originates”: e.g., lines 32, 441, 616 (ascending process (coronoid process) that serves as insertion …), 707 (projects dorsomedially where the suparcualis rectus II attaches …); 776 (… end expanded where the pectoral muscles attach.); 790; 805, 806; 825; 827; 869; 902, 904, 906; 914, 916; 918
- throughout text capitalize Fig., e.g., lines 662, 720, 1017, 1047, 1048 (and add space between E, and H), 1055, 1056, 1065, 1066

Changes by line: SPECIFIC
- line 55: The family Hynobiidae includes 66-67 species …
- lines 66-81: This paragraph can be shortened as information are repeated.
- lines 96-97: How does the description of the osteology contribute to the protection of the species? Please, explain.
- lines 107-113: Delete the sentence (starts with “Two”) – this information is not relevant for this paper.
- line 151: Reilly & Lauder
- line 152: Shubin & Wake
- line 156: add space between number and ‘in’; introduce abbreviation CIB after Chengdu Institute of Biology  use CIB in lines 162, 165
- line 161: add comma after 2014
- line 245: “… londongensis”
- line 259: move ‘characters’ to end of the sentence.
- line 288: refer to one of your figures here
- line 307: CN V1?
- line 328: of the skull roof (not in …)
- line 364: add space after posteromedially
- line 369: … the lacrimal extends anteriorly over the border of …  change wording accordingly throughout text
o 978, 979
- lines 384-385: … through which the superior alveolar branch of the trigeminal nerve (CN V2) and its associated blood vessels pass as in other salamanders (Francis, 1934).
- lines 404-405: delete both commas in this sentence
- line 412: The quadrate, as a principal element of the supsensorium, extends …
- line 427: add comma after ‘(Lebedkina, 1964)’
- line 448: Can you find a different term for ‘propping’? I am not sure what you want to say by this.
- lines 461-2: However, B. karlschmidti …
- line 467: homologous origin or a derived state …
- 474: Those fenestra …
- 478: this should be 14482 instead of 14382? and 14509 has no gills – is it neotenic?
- 516-7: ramus ventralis of trigeminal nerve (CN V1, 2, or 3?)
- 520: The parasphenoid is a dermal bone which forms …
- 529: sort the numbers of the CIB specimens (14482 shouldn’t be the last one)
- 532: … but has an articulation with …
- 559: … stapedial foramen in specimens observed by him.
- 567-570: move paragraph to line 547
- 582-3: … bears the occipital condyle posteriorly which articulates with the cotyle of the atlas.
- 591-2: rephrase this sentence: a bone is not supplying things (arteries do supply): e.g. “The ventral border of the foramen magnum is largely formed by the posteromedian process of the parasphenoid.”
- 596: add comma after ‘angular’
- 599: bone of the mandible
- 603: innervating both the skin and the muscle (arteries supply, nerves innervate)
- 606: add comma after ‘ventral’
- 625-6: In dorsal view, the anterior process of the articular extends between …
- 641: ossified during metamorphosis
- 644: add comma after ‘maxilla’ and delete ‘and’
- 645: … vomer, and dentary.
- 655: There are 20-25 teeth on the dentary.
- 665: add ‘our’ before CT-scanned images  it is actually possible to see cartilages in CT scans if the scanning parameters can be adjusted (also depends on size of specimens), see for example:
o Figure 3. Anatomic section (A) and CT image (B) of the canine larynx at the section level 10 of the figure 1. : http://mingaonline.uach.cl/scielo.php?pid=S0301-732X2010000100013&script=sci_arttext
- 680: delete ‘(see discussion below)’ as your results are mixed with discussion – if you want to add something you can say ‘(see below)’
- 692: add comma after ‘(Fei et al., 2006)’
- 708: Ceratobranchials III and IV
- 726: … from all B. londongensis specimens studied here, …
- 728: … caudal (tail) vertebrae
- 733: … widened anteriorly, where it bears a pair of …
- 735: … articulates with the exopccipitals at …
- 737: … pair of foramina through which the first pair of spinal nerves pass. The atlas has no …
- 745: … All presacral vertebrae except the atlas articulate with free ribs. (Ribs are not part of vertebrae)
- 747-8: refer to a figure or add one; also are the ribs bifurcated or do they appear bifurcated because of the uncinate process?
- 758: elongated
- 761: CIB 14485 is the only …
- 765-6: In fully grown adults, 28-30 vertebrae follow the caudosacral series.
- 768-9: All caudal vertebrae, except the posterior-most two or three, ventrally bear a haemal arch, through which the haemal vessels pass.
- 779: cite Table 1 instead of listing CIB specimens
- 785: What do you man with ‘received’? Articulates?
- 792: delete ‘coracoid’ ? Does it really articulate with the humerus?
- 794: … deeply notched where the crista ventralis of the humerus fits …
- 799-804: but you can relate it to size?
- 813: refer to figure? move specimens to end of the sentence (814)
- 820: … two faces for articulation with …
- 830: … radius and ulna are also ossified, …
- 844: add comma after ‘distally’
- 845: delete comma after ‘proximally’
- 860: delete comma after ‘acetabulum’
- 861-2: … vertical position, while the iliac blade is posteriorly slightly inclined.
- 873: delete ‘several’
- 877: refer to Fig. 10
- 883-4: To our knowledge, there are no publications which figured or …
- 887: delete ‘ypsiloid’ (you already have ‘y-shaped’)
- 899: femur (not humerus)
- 909: delete ‘obviously’
- 910: … larger than the fibula condyle.
- 920: … to articulate with the tibia.
- 927: add comma after ‘proximally’
- 928: Distal tarsals 3 and 4 are similar size and both are …
- 929: delete ‘obviously’; Distal tarsal 5 is missing, what is likely related to the loss of digit 5.
- 931-2: delete sentence as paragraph below (935-42) states the same.
- 957: The fibulare is enlarged and roughly …
- 961: Change heading to Summary as you discuss throughout your results
- 980: ‘absent owing to intervention by the lacrimal’ – Please rephrase.
- 991: … four to eight vomerine teeth, …
- 991: add comma after ‘orientated’
- 996: … 18 presacral vertebrae; … (same for line 1139)
- 997: … short with a prominent …
- 1004-6: delete sentence; you already stated that in Materials & Methods
- 1019 & 1036 & 1146 & 1467: ceratobranchials III and IV
- 1019: With regard to …
- 1020: delete ‘so-called’
- 1028: With respect to …
- 1033: Evidently, ossification …
- 1038: … retains a remnant …
- 1050: … ossified in both species.
- 1055: …; however, Xiong et al., 2013b: Fig. 1G shows …
- 1063: … has yet to be recorded.
- 1081-3: delete ‘in the literature’ and add citation at the end of this sentence
- 1099: … have a large radiale, …
- 1108: … Batrachuperus, several other …
- 1111: Shubin & Wake
- 1117-8: … plesiomorphic feature because it is widely distributed in the family Hynobiidae. However, how many taxa …
- 1125-31: rephrase
- 1158-63: add locations: City, USA / CAN / China …
- 1166-74: that might be journal requirement, but the information is repeated
- 1150: delete ‘(B)’: (B) CIB 14481; (C) CIB 14482, with an incomplete resorption …
- 1453: (A) holotype CIB 65I0013/…
- 1458-9: (A) holotype CIB 65I0013/…
- 1458-61: Figure 4 & 1481-3 Figure 9: add the ‘notes’ where they belong and delete ‘note’ see example above (line 1150)

---

## Round 0.2 · Minor Revisions

The manuscript has been revised substantially, although I see that the authors have chosen not to follow some of the suggestions raised by the reviewers. At least for the wording and phrasing, I urge the authors to follow the suggestion by reviewer #2 (see his/her comments on wording for attachments and insertions) as the current wordings are not correct in the field of anatomy.

The authors seem to have a strong opinion on not citing Marjanovic & Laurin (2014). PeerJ is a new platform for open science, and authors are encouraged to expose their opinions and original thoughts in this journal. In this case, I would like you to make the peer-review history of your submission 'open' (i.e. the submission, the reviews, the rebuttal etc are made publicly available upon publication using PeerJ's 'Open Peer Review' functionality). Given the disagreements that are apparent, I believe this extra transparency will explain the review process and the author's reasoning to the readers. I will be happy to accept this manuscript if the authors agree to make this review open.

·

Basic reporting

The authors have now responded to most of my comments. In my opinion, the minor, stylistic revisions are fine, but the reasons they provide not to develop their embryonic discussion further, or not implement other more substantial changes, are not convincing.

To take only one example, in their rebuttal letter, the authors state “We did not cited Marjanovic & Laurin (2014) because their updated paleontological timetree of lissamphibians provided no specific estimations of Hynobiidae-Cryptobranchidae splitting time.” This is wrong and surprising, and even though the authors revised their sentence regarding the divergence date between hynobiids and cryptobranchids, it is still incompatible with the timetree of Marjanovic & Laurin (2014) because the authors now state: “Recent analyses of the nuclear exon and mitochondrial genome estimate the Cryptobranchidae-Hynobiidae split as ~150 Ma and the origin of crown-group hynobiids as ~125 Ma (Zheng et al., 2011: fig. 3).” According to Marjanovic & Laurin’s (2014) timetree, there are pan-hynoobiids and pan-cryptobranchids from the Bathonian (166-168 Ma). There is a clear incompatibility between these estimates (for once, the molecular estimates look too young), and I am extremely surprised that they authors don’t see this and don’t wish to comment on this. If new dating of the formations of the relevant fossils falsify the timetree proposed by Marjanovic & Laurin (2014), or if the ideas about the affinities of these fossils have changed, it would be a service to the community to indicate this. The timetree indicates that the divergence dates at least from the Bathovian (Middle Jurassic).

The authors should at least rework this point, I think. For the rest, given that I view my role as providing advice, not forcing anybody to do something against their will, so I will not insist further. Some authors are keen on extracting as much impact out of their results as possible, whereas others shy away from drawing inferences based on their findings, and thus, let others exploit their results. The authors appear to fall into this second category ; so be it. They are the ones who will suffer from their academic shyness, and others may reap the rewards of making interesting inferences from these findings, if they happen to notice them (of course, this is far from certain).

Experimental design

Paleontology is not an experimental science, like the rest of morphology and systematics. It is a historical and descriptive science. Thus, most of the points mentioned in the form do not apply.

Validity of the findings

Findings seem valid. My only comment is still that broader relevance the findings needs to be highlighted better by more thorough discussions of the most interesting findings. The current version is clearly under-developed, on this front, but given that the authors are uninterested in improving this aspect of the paper, I have nothing to add.

Additional comments

I have nothing to add. See above.

·

Basic reporting

no comment

Experimental design

no comment

Validity of the findings

no comment

Additional comments

The MS has improved and while I recognize that many suggestions from both reviewers were included, others were not. Most of the not-included ones are well justified in the ‘Rebuttal-letter’. However, one particular suggested change was statet as done but turned out to be not done.
In my first review, I wrote: “throughout text change phrasings like “structure FOR attachment (origin, insertion) of muscle xy” to “structure, onto which muscle xy attaches/inserts” or to “structure, from which muscle xy originates”: e.g., lines 32, 441, 616 (ascending process (coronoid process) that serves as insertion …), 707 (projects dorsomedially where the suparcualis[subarcualis] rectus II attaches …); 776 (… end expanded where the pectoral muscles attach.); 790; 805, 806; 825; 827; 869; 902, 904, 906; 914, 916; 918”
The authors replied: “—Suggested changes have been made throughout our manuscript text.”
But now in the V1 the authors still write: “Proximally on its extensor aspect, the crista dorsalis (dorsal crest) bears a prominent triangular or even knob-like projection, for insertion of the M. subscapularis (Francis, 1934). Proximally on the flexor aspect, the crista ventralis (ventral crest) is a large triangular process, projecting anteroventrally for attachment of the M. pectoralis and M. supra-coracoideus (Francis, 1934) and merging with the head of the humerus (or a facet in some specimens; see below).”

The phrases “For attachment” or “for insertion” can still be found numerous times in the text, but in general this is a form of description that should be avoided – e.g., facets are not developed for the attachment of condyles – both develop & evolve together, processes are not developed for muscles, …. Simply, search the document for those phrases and you will see where it should be corrected. However, if the Editor is fine with the phrase/non-correction, I won’t object.

---

## Round 0.3 · accepted · Accept

I have reviewed your revision, and it is a pleasure to inform you that your manuscript is accepted for publication. Thank you for your wonderful contribution. Once again, just please be assured that you make the review process open.